# PREDICATE HIERARCHIES IMPROVE
# FEW-SHOT STATE CLASSIFICATION

**Emily Jin**[*]
Stanford University

**Joy Hsu**[*]
Stanford University

**Jiajun Wu**
Stanford University

## ABSTRACT

State classification of objects and their relations is core to many long-horizon tasks, particularly in robot planning and manipulation. However, the combinatorial explosion of possible object-predicate combinations, coupled with the need to adapt to novel real-world environments, makes it a desideratum for state classification models to generalize to novel queries with few examples. To this end, we propose PHIER, which leverages *predicate hierarchies* to generalize effectively in few-shot scenarios. PHIER uses an object-centric scene encoder, self-supervised losses that infer semantic relations between predicates, and a hyperbolic distance metric that captures hierarchical structure; it learns a structured latent space of image-predicate pairs that guides reasoning over state classification queries. We evaluate PHIER in the CALVIN and BEHAVIOR robotic environments and show that PHIER significantly outperforms existing methods in few-shot, out-of-distribution state classification, and demonstrates strong zero- and few-shot generalization from simulated to real-world tasks. Our results demonstrate that leveraging predicate hierarchies improves performance on state classification tasks with limited data.

## 1 INTRODUCTION

State classification of objects and relations is essential for a plethora of tasks, from scene understanding to robot planning and manipulation (Migimatsu & Bohg, 2022; Chen et al., 2024). Many such long-horizon tasks require accurate and varied state predictions for entities in scenes. For example, planning for "setting up the table" requires classifying whether the `cup` is `NextTo` the `plate`, whether the `utensils` are `OnTop` of the `table`, and whether the `microwave` is `Open`. The goal of state classification is to precisely answer such queries about specific entities in an image, and determine whether they satisfy particular conditions across a range of attributes and relations.

However, the combinatorial space of objects (e.g., `cup`, `plate`, `microwave`) and predicates (e.g., `NextTo`, `OnTop`, `Open`) gives rise to an explosion of possible object-predicate combinations that is intractable to obtain corresponding training data for. In addition, real-world systems operating in dynamic environments must generalize to queries with novel predicates, often with only a few examples (Bendale & Boult, 2015; Joseph et al., 2021; Ha & Song, 2022). For instance, we may

---

[*]Equal contribution

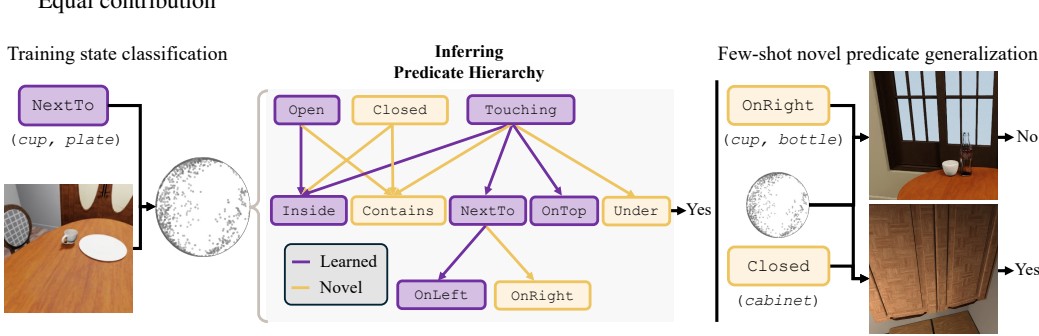

Figure 1: PHIER improves few-shot state classification, by encoding a *predicate hierarchy* in joint image-predicate latent space. By encouraging such structured representations to emerge, PHIER enables strong few-shot generalization to novel predicates with few examples.

not only want to classify our trained query of whether the `cup` is `NextTo` the `plate`, but more specifically, whether the `cup` is `OnRight` of the `bottle`, and whether the `cabinet` is `Closed` (See Figure 1). Hence, an essential but difficult consideration for state classification models is to quickly learn to adapt to out-of-distribution queries. However, current supervised methods struggle with few-shot state classification, and pretrained large vision language models fail to accurately answer challenging spatial relationship queries in robotics environments.

To this end, we propose PHIER, a state classification model that leverages the hierarchical structure between predicates to few-shot generalize to novel queries. At the core of PHIER is an image-predicate latent space trained to encode the relationship between pairwise predicates (See Figure 1). Let us consider the predicates, `OnRight` and `OnLeft`—while they describe opposite spatial relationships between objects, they are closely related semantically, as assessing them relies on the same underlying features. PHIER enforces image-predicate representations conditioned on these predicates to lie closer to one another. In addition, there exist predicate pairs with more complex relationships, such as `OnRight` and `NextTo`. We see that `OnRight` is a more specific case of `NextTo`—verifying `OnRight` involves recognizing whether the objects are `NextTo` each other. Features relevant to the higher-level predicate `NextTo` are therefore useful for reasoning about the lower-level predicate `OnRight`. PHIER encodes this *predicate hierarchy* to allow generalizable state classification.

Such a predicate hierarchy can be used to quickly adapt to unseen object-predicate pairs. For example, when combining a learned predicate `NextTo` with the objects `apple` and `microwave`, PHIER uses a learned embedding of the predicate to control the features extracted from the image depicting an apple and microwave. Notably, for a more challenging state with a novel predicate, such as `OnRight`, the model can still adapt by drawing on its relationship to `OnLeft` and `NextTo` to learn relevant features from limited data. PHIER leverages pairwise hierarchical relations between predicates to efficiently generalize to out-of-distribution predicates with few examples.

To perform state classification, PHIER first localizes relevant objects in the input image based on a given query, then leverages an inferred predicate hierarchy to structure its reasoning over the scene. PHIER processes objects and predicates separately, learning to map object representations conditioned on the predicate into an image-predicate latent space. The hierarchical structure of predicates is learned through self-supervised losses based on the relationships between predicates (e.g., `OnRight` and `NextTo`). We use a large language model (LLM) to infer the pairwise predicate relations from language. However, explicit hierarchies inherently assume some discretization over predicates, which does not align with the continuous nature of representations used in deep learning (Nickel & Kiela, 2017; Ganea et al., 2018). To address this, we propose using a hyperbolic distance metric to encode the hierarchical structure of predicates in a continuous manner, enabling PHIER to effectively incorporate tree-like structure.

We evaluate PHIER on the state classification task in two robotics environments, CALVIN (Mees et al., 2022) and BEHAVIOR (Li et al., 2023a). Beyond the standard test settings, we focus on few-shot, out-of-distribution tasks involving unseen object-predicate combinations and novel predicates. Our results show that PHIER significantly outperforms existing methods, including both supervised approaches trained on the same amount of data and inference-only vision-language models (VLMs) trained on large corpora of real-world examples. PHIER improves upon the top-performing prior work in out-of-distribution tasks by 22.5 percent points on CALVIN and 8.3 percent points on BEHAVIOR. Notably, trained solely on simulated data, PHIER also outperforms supervised baselines on zero- and few-shot generalization to real-world state classification tasks by 7 percent points and 10 percent points respectively. Overall, we see PHIER as a promising solution to few-shot state classification, enabling generalization by leveraging representations grounded in predicate hierarchies.

In summary, our contributions are the following:

- We introduce PHIER, a state classification model that encodes inferred predicate hierarchies in its latent space, enabling generalization to unseen object-predicate combinations and novel predicates with few examples.
- We propose learning the predicate hierarchy through an object-centric scene encoder, self-supervised losses that encourage pairwise predicate relations, and a hyperbolic distance metric to model the hierarchical nature of predicates in continuous space.
- We validate PHIER's performance in few-shot, out-of-distribution generalization, and zero- and few-shot real-world transfer across three state classification datasets. PHIER significantly outperforms both supervised baselines and large pretrained VLMs.

## 2 RELATED WORK

**State classification.** The ability to understand object states and relationships is essential for a wide range of tasks in computer vision and robotics, including scene understanding, robotic manipulation, and high-level planning (Yao et al., 2018; Yuan et al., 2022). Earlier works that focus on a similar task of visual relationship detection learn to extract object-centric representations from raw images and make predictions based on them (Gkioxari et al., 2018; Yao et al., 2018; Ma et al., 2022; Yuan et al., 2022). A more recent approach by Yuan et al. (2022) specifically addresses state classification by extracting object-centric embeddings from RGB images and feeding them into trained networks to classify a set of predefined predicates. However, their approach relies on input images of the objects of interest and is limited to predicates seen during training, as it requires a separate network for each predicate. In contrast, our method can few-shot generalize to unseen predicates given only the input scene and query, without any additional annotations or specific object images.

Recent advancements in robotics simulators have additionally enabled scalable training of state classification in simulation and subsequent transfer to real-world settings. Simulators such as CALVIN (Mees et al., 2022) and BEHAVIOR (Li et al., 2023a) offer varying levels of realism and are widely used in the robotics community to generate large and diverse datasets, supporting data generation across a broad range of states (Ge et al., 2024). We train our model on such simulators and show that, compared to prior works, PHIER is significantly more effective at zero- and few-shot generalization to real-world state classification tasks.

**Question-answering approaches.** Several key advancements in visual question answering (VQA) have been driven by the development of pretrained large vision-language models (VLMs). These models are trained on extensive image and text data, leading to a unified visual-language representation that can be applied to perform various downstream tasks (Shen et al., 2021; Li et al., 2023b; OpenAI, 2023). Approaches such as Viper-GPT (Surís et al., 2023) further leverage the power of foundation models by composing LLMs to generate programs that pretrained VLMs can execute. However, despite their strong general-purpose capabilities, these models still struggle with questions involving visual and spatial relationships (Tong et al., 2024).

On the other hand, a separate class of VQA methods are models trained directly for the VQA task. These approaches follow a general framework of extracting visual and textual features, combining them into a multimodal representation, and learning to predict the answer. Convolutional neural networks (CNNs) and transformers are widely used for feature extraction, with various techniques for fusing the features. FiLM (Perez et al., 2018) is an early, modular approach that applies a feature-wise transformation to condition the image features on the text features, while BUTD (Anderson et al., 2018) and Re-Attention (Guo et al., 2020) are representative attention-based methods that localize important regions based on the question. Furthermore, many approaches, including modular, graph-based, or neurosymbolic approaches, introduce more explicit reasoning to better model the relationships and interactions between objects (Andreas et al., 2016; Yi et al., 2018; Ma et al., 2022; Nguyen et al., 2022; Wang et al., 2023). Our work lies in this latter class of methods, designed and trained for state classification. In contrast to prior works, we not only encode visual features and textual features of predicates but also learn to capture the inherent predicate hierarchy in a joint image-predicate latent space.

**Hyperbolic representations for hierarchy.** In recent years, several works have explored the benefits of using hyperbolic space to model hierarchical relationships, as it is particularly well-suited for capturing hierarchical structures (Ganea et al., 2018; Nickel & Kiela, 2018). In computer vision, prior works have focused on learning hyperbolic representations for image embeddings, demonstrating their effectiveness across various tasks such as semantic segmentation and object recognition (Khrulkov et al., 2020; Liu et al., 2020; Atigh et al., 2022; Ermolov et al., 2022; Ge et al., 2023). Hyperbolic embeddings allow models to naturally represent hierarchical relationships between various entities, such as objects and scenes or different categories, leading to improved performance and generalization in such tasks (Weng et al., 2021; Ge et al., 2023). Several approaches further incorporate self-supervised learning to learn the underlying hierarchical structure without the need for explicit labels, instead leveraging proxy tasks such as contrastive learning (Hsu et al., 2021; Ge et al., 2023; Yue et al., 2023). Recently, Desai et al. (2023) proposed a contrastive method to learn joint representations of vision and language in hyperbolic space, yielding a representation space that captures the underlying structure between images and text. Inspired by these works, PHIER learns a predicate hierarchy in hyperbolic space informed by language and the pairwise relations between predicates, and uses it to conduct few-shot generalization to unseen state classification queries.

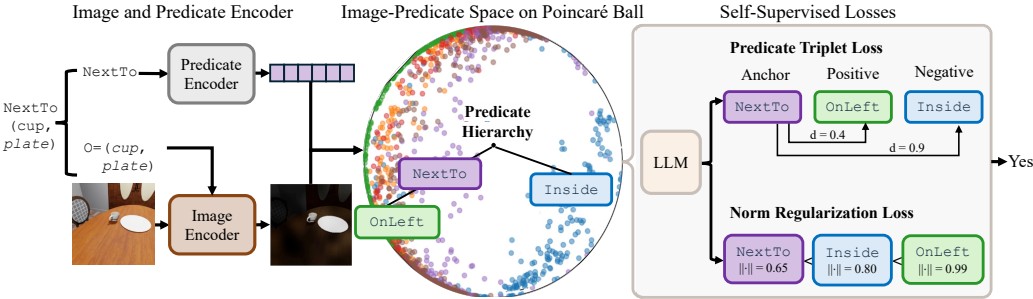

Figure 2: PHIER consists of three main components. The first are disentangled image and predicate encoders, which separately extract features based on the objects and predicates in the state classification query. The second is a self-supervised learning process that injects explicit knowledge of pairwise predicate relations into the image-predicate latent space. The third is the use of a hyperbolic distance metric and encoder to encourage encoding of the inferred predicate hierarchy. Together, these components enable few-shot generalization to unseen object-predicate pairs and novel predicates.

## 3 METHOD

In this section, we describe PHIER, our model for generalizable state classification. We define PHIER's task as a binary state classification problem: given a 2D RGB image and a text query describing a state (e.g., NextTo(cup, plate), the goal is to predict whether the predicate NextTo holds True or False for the objects, cup and plate, in the input image. PHIER enables efficient generalization to unseen predicates, by way of inferring a structured latent space of image-predicate pairs to perform reasoning over (See Figure 2). At the core of our method is a predicate hierarchy that captures the semantic relationships between predicates. This hierarchical structure is inferred through three key components:

1. An object-centric scene encoder that localizes regions corresponding to relevant objects;
2. Self-supervised losses that inject pairwise relations of predicates into the latent space;
3. A hyperbolic distance metric and an encoder that model hierarchical representations.

In Section 3.1, we describe PHIER's base architecture of object and predicate conditioning. In Section 3.2, we introduce self-supervised losses that encourage representations to cluster based on pairwise predicate relationships. In Section 3.3, we detail how PHIER learns hierarchical representations by embedding representations in hyperbolic space.

### 3.1 OBJECT-CENTRIC IMAGE ENCODER

PHIER extracts an object-centric scene representation by conditioning the input image $\mathbf{I}$ on the query. Assume the input query is NextTo(cup, plate). PHIER first parses it into its objects $\mathcal{O} = \{cup, plate\}$ and predicate $\mathcal{P} = NextTo$. Then, PHIER separately conditions the latent space on both of these components (See Figure 2). By disentangling the full state classification query, we ensure that PHIER faithfully identifies the relevant entities; it then learns to focus on their key features for the given predicate's classification.

**Object conditioning.** The goal of object conditioning is to localize regions of the image that correspond to the relevant objects in the query. This allows PHIER to focus on the objects of interest for predicting the downstream predicate condition. To encode an input image $\mathbf{I}$ on the objects $\mathcal{O}$, our encoder $E_{\text{img}}$ generates an object-conditioned image mask $\mathcal{M}(\mathbf{I}, \mathcal{O})$ that highlights image regions based on the relevant entities, following Zhou et al. (2022). At a high level, we extract features for the image and object texts, align the image features with each object text's features to generate individual object masks, and then encode the image based on these masks. Concretely, for each object $\mathbf{o} \in \mathcal{O}$, we use a CLIP image encoder $\mathcal{V}$ and text encoder $\mathcal{T}$ (Radford et al., 2021) to obtain image features $\mathcal{V}(\mathbf{I}) \in \mathbb{R}_1^D$ and object text features $\mathcal{T}(\mathbf{o}) \in \mathbb{R}_2^D$, where $D_1, D_2$ correspond to the visual and language embedding dimensions, respectively.

To align the image and text features, we first project the image features into the same space as the text features. In order to preserve the spatial structure of the image features, we initialize a $1 \times 1$ convolution layer $\mathcal{C}$ with weights from the last linear layer of the CLIP attention pooling mechanism and apply this to the image features. Then, we apply a convolution $\otimes$ on the image features, using the text features as filters. This directly aligns image regions with the object text, highlighting the parts of the image that are the most relevant to the query. Our resulting object mask $\mathcal{M}(\mathbf{I}, \mathbf{o})$ is defined as

$$\mathcal{M}(\mathbf{I}, \mathbf{o}) = \mathcal{C}(\mathcal{V}(\mathbf{I})) \otimes \mathcal{T}(\mathbf{o}).$$

To obtain the final image mask, we upsample each object mask to the original image dimensions, normalize the values to the range $[0, 1]$, and take the element-wise $\max$ across the individual object masks. We then apply the final mask to our image using a Hadamard product $\odot$ and subsequently encode the masked image using a pretrained vision transformer (ViT) encoder (Dosovitskiy, 2020) $E_{\text{ViT}}$ as follows:

$$E_{\text{img}}(\mathbf{I}, \mathcal{O}) = E_{\text{ViT}}(\mathbf{I} \odot \max_{o \in \mathcal{O}}(\text{norm}(\text{upsample}(\mathcal{M}(\mathbf{I}, \mathbf{o}))))).$$

This ensures that the representation encodes all relevant regions based on the specified objects.

**Predicate conditioning.** Next, PHIER conditions the latent representation $E_{\text{img}}(\mathbf{I}, \mathcal{O})$ on the text predicate $\mathcal{P}$, to focus on features essential to the downstream classification task. To do so, we encode the predicate text $P$ using a pre-trained BERT encoder $E_{\text{text}}$ (Devlin, 2018) and concatenate this with the masked image, resulting in a final object-centric scene representation:

$$E(\mathbf{I}, \mathcal{O}, \mathcal{P}) = \text{concat}(E_{\text{img}}(\mathbf{I}, \mathcal{O}), E_{\text{text}}(\mathcal{P})),$$

which incorporates both object features and context relevant to the predicate.

## 3.2 Self-Supervised Learning

PHIER learns a joint image-predicate space that encodes an inferred *predicate hierarchy* with self-supervised losses. In particular, a specific predicate such as `OnLeft` is encouraged to be close to a related, more general predicate such as `NextTo`, as features important to closer predicates tend to be more alike. At the same time, `OnLeft` should have a larger norm than `NextTo`, to reflect its lower position in the hierarchy. This ensures that the features that are learned to be relevant to higher-level predicates remain useful when reasoning about their lower-level children.

We introduce two self-supervised losses to encourage such pairwise relationships: a predicate triplet loss based on the similarity between predicates and a norm regularization loss based on the hierarchy between predicates. For both losses, we sample triplets with unique corresponding predicates and then extract knowledge from an LLM to determine the underlying relationships between the predicates.

**Predicate triplet loss.** Given a predicate triplet with an anchor $a$, positive $p$, and negative $n$ sample, the triplet loss encourages the distance between the anchor and negative predicate to be at least some margin $\lambda$ greater than the distance between the anchor and positive predicate. In Figure 2, we see that `NextTo` is the anchor predicate, `OnLeft` is the positive sample, and `Inside` is the negative sample. The proper assignment of the samples is critical, as it directly affects how faithfully the model learns the relationship between predicate pairs. To determine the anchor, positive, and negative sample for any given predicate triplet, we query an LLM based on the semantic meanings and scene details described by each query. More concretely, for each triplet, we prompt the LLM to assess the underlying relationships between the predicates. One predicate in the triplet is randomly chosen as the anchor. The LLM is asked to determine which of the other two predicates is more similar to the anchor. The anchor and the more similar predicate form a positive pair, while the anchor and the less similar predicate form a negative pair. We provide the prompt template provided in Appendix D. By extracting knowledge from an LLM, we leverage the LLM's explicit and extensive understanding of predicate relationships to produce meaningful triplets and guide the model toward a semantically rich image-predicate latent space.

Our formulation of the triplet loss is based on the distance between representations:

$$\mathcal{L}_{\text{triplet}, \lambda}(a, p, n) := \max(0, \mathbf{d}(a, p) - \mathbf{d}(a, n) + \lambda),$$

where $\mathbf{d}$ is the distance metric used in the latent space. We describe our choice of a hyperbolic distance metric in the following section. With this loss and chosen distance metric, the resulting representation space captures the similarity between predicates via their distance in the latent space.

**Norm regularization loss.** While the triplet loss enforces similarity between related predicates, the norm regularization term introduces hierarchical structure to the latent space. We leverage the LLM's strong semantic understanding of predicates to infer the hierarchy among a triplet, by ranking the predicates based on specificity. Specificity depends on several factors, including the variety and number of objects required by the state, the important features of the objects and relationships between the objects, the level of detail required by the state, and the semantic meaning of each description. The prompt template is provided in Appendix D.

Given a ranking of three predicates $a, b, c$ from least to most specific, the regularization loss encourages the norm to increase by at least some margin $\gamma$ as the specificity increases:

$$\mathcal{L}_{\text{reg},\gamma}(a, b, c) := \max(0, ||b|| - ||a|| + \gamma + \max(0, ||c|| - ||b|| + \gamma).$$

Intuitively, the regularization loss ensures that the norms of the representations reflect the implicit hierarchy between predicates.

Together, the predicate triplet loss and norm regularization loss encourage a semantically rich and structured representation space. The predicate triplet loss captures the similarity between predicates by enforcing appropriate distances between related predicates, while the norm regularization loss ensures that the hierarchical relationships are reflected in the norm of the representations. By leveraging the explicit knowledge of an LLM to infer both the triplet assignments and hierarchy ranking, our approach ensures that PHIER's learned representations align with the underlying predicate ontology.

### 3.3 Hyperbolic Image-Predicate Latent Space

Finally, PHIER effectively learns this inferred hierarchy of predicates through *hyperbolic* representations. While PHIER's self-supervised losses inject semantic knowledge of predicates into its representations, PHIER further encodes the hierarchical nature of the predicates in hyperbolic space. In hyperbolic space, we can more easily visualize these relationships forming a natural predicate hierarchy. In Figure 2, we visualize a learned predicate hierarchy in PHIER's latent space.

Hyperbolic space is a non-Euclidean space characterized by constant negative curvature, which allows hierarchical structure to be easily embedded. Due to hyperbolic space's curvature, the area of a disc increases exponentially with its radius, analogous to the exponential branching of trees. This property makes hyperbolic space well-suited for modeling hierarchies, as it provides a continuous representation of discrete trees. Furthermore, hyperbolic space is differentiable, making it easy to integrate with our model. Hence, we propose using a hyperbolic distance metric for our predicate triplet loss and norm regularization loss to more effectively encode the predicate hierarchy. These hierarchical representations enable PHIER to generalize effectively to novel predicates, by inferring their representations based on their relationships to learned predicates in the hyperbolic latent space.

**Poincaré ball model.** In this work, we use the Poincaré ball model of hyperbolic space. The Poincaré ball is an open $d$-dimensional ball of radius 1, equipped with the metric tensor $g_{\mathbf{p}} = (\lambda_x)^2 g_{\mathbf{e}}$. Here, $|| \cdot ||$ is the Euclidean norm, $\lambda_x = \frac{2}{1-||x||^2}$ is the conformal factor, and $g_e$ is the Euclidean metric tensor (i.e., the Euclidean dot product). This induces the distance $\mathbf{d_p}$ between two points $x, y$ on the Poincaré ball as

$$\mathbf{d_p}(x, y) = \cosh^{-1}\left(1 + 2\frac{||x - y||^2}{(1 - ||x||^2)(1 - ||y||^2)}\right).$$

On the Poincaré ball, the distance between two points captures the degree of similarity between them, while the relative norm of two points reflects their hierarchical structure. Thus, the Poincaré ball is a suitable space to represent the underlying predicate hierarchy, and PHIER's self-supervised losses use such metrics to embed image-predicates onto the Poincaré ball.

**Hyperbolic encoder.** To obtain the hyperbolic image-predicate representation, we use the exponential map to lift the representation from Euclidean space onto the Poincaré ball and pass it through a small hyperbolic linear network (Ganea et al., 2018). For more details on these functions, we refer the reader to Appendix C.

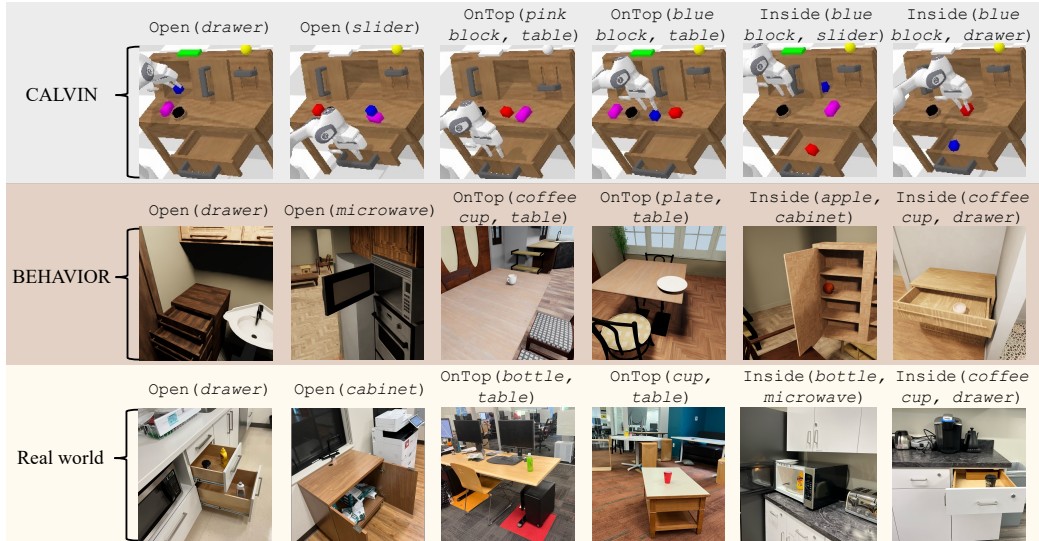

Figure 3: Examples of state classification tasks from CALVIN and BEHAVIOR. The datasets span a range of visual realism and complexity.

## 3.4 TRAINING LOSS

After the hierarchical representation is learned in hyperbolic space, we apply the logarithmic map to project it back to Euclidean space, where it is passed through a small MLP for state classification. We train PHIER with a binary cross entropy loss based on the ground truth labels (True or False), along with the predicate triplet loss and the norm regularization loss. Our overall loss is defined as $\mathcal{L}_{\text{total}} := \mathcal{L}_{\text{sup}} + \alpha \mathcal{L}_{\text{triplet},\lambda} + \beta \mathcal{L}_{\text{reg},\gamma}$, where $\alpha$, $\beta$ are coefficients that control the strength of the triplet and regularization losses, and $\lambda$, $\gamma$ are the margins for the two losses, respectively.

## 4 DATASET

We evaluate PHIER on established robotics environments, with three state classification datasets designed to test the following key aspects of performance: a faithful understanding of entities and relations between them, few-shot generalization to out-of-distribution queries, and zero- and few-shot transfer to a real-world setting. See Figure 3 for examples from each of the environments.

**Simulator dataset generation.** In order to evaluate our method's state classification performance on robotic environments, we generate datasets of varying levels of realism with two widely used robotics simulators, CALVIN (Mees et al., 2022) and BEHAVIOR (Li et al., 2023a). Both are known for their ease of use and customizability, allowing us to generate diverse data for various states with different objects under various lighting, camera angle, and object pose conditions. As shown in Figure 3, CALVIN is visually simplistic while BEHAVIOR is more realistic and complex. By evaluating on data from these two simulators, we assess how well various methods understand the semantics of different predicates.

**Simulator dataset details.** To evaluate the effectiveness of our inferred abstraction hierarchy, we define sets of in-distribution and out-of-distribution states, featuring both unary and binary relations. The out-of-distribution states involve unseen predicate-object combinations and novel predicates. See Appendix E for the states in each dataset. We train on a balanced dataset of 200 examples (100 True, 100 False) for each in-distribution state. We then evaluate on balanced test sets of 50 examples for each state under both in-distribution and out-of-distribution settings.

**Real-world dataset.** In addition, we evaluate on BEHAVIOR Vision Suite (Ge et al., 2024) (see Figure 3), a complex real-world benchmark that consists of diverse scenes and distractor objects. Specifically, compared to our train data, this one consists of 10 unseen combinations and 10 novel predicates, with 337 total examples. We use this dataset to test our method's ability to perform zero- and few-shot real-world transfer after training on simulated datasets alone.

Table 1: Comparison of PHIER to prior works on CALVIN and BEHAVIOR datasets. We compare against trained models (T) and inference-only models pretrained on large-scale data (I). In this table, we report in-distribution (ID) test accuracy, out-of-distibution (OOD) test accuracy, and the difference in performance between the two test sets (ID-OOD). PHIER significantly outperforms prior work in the OOD setting; it is also the only method that performs similarly in ID and OOD settings.

| | | CALVIN | | | BEHAVIOR | | |
|---|---|---|---|---|---|---|---|
| | | ID ↑ | OOD ↑ | ID-OOD ↓ | ID ↑ | OOD ↑ | ID-OOD ↓ |
| PHIER (Ours) | T | 0.945 | **0.899** | **0.046** | 0.859 | **0.820** | **0.039** |
| Re-Attention (Guo et al., 2020) | T | **0.959** | 0.674 | 0.285 | 0.828 | 0.652 | 0.176 |
| CoarseFine (Nguyen et al., 2022) | T | 0.878 | 0.624 | 0.254 | 0.766 | 0.636 | 0.130 |
| BUTD (Anderson et al., 2018) | T | 0.898 | 0.585 | 0.313 | 0.808 | 0.712 | 0.096 |
| RelViT (Ma et al., 2022) | T | 0.688 | 0.563 | 0.125 | **0.866** | 0.737 | 0.129 |
| CLIP (Shen et al., 2021) | T | 0.937 | 0.546 | 0.391 | 0.722 | 0.632 | 0.090 |
| FiLM (Perez et al., 2018) | T | 0.798 | 0.489 | 0.309 | 0.753 | 0.583 | 0.170 |
| SORNet (Yuan et al., 2022) | T | 0.943 | – | – | 0.773 | – | – |
| GPT-4V (OpenAI, 2023) | I | 0.587 | 0.563 | 0.024 | 0.661 | 0.706 | −0.045 |
| BLIP-2 (Li et al., 2023b) | I | 0.553 | 0.556 | −0.003 | 0.554 | 0.535 | 0.019 |
| ViperGPT (Surís et al., 2023) | I | 0.466 | 0.475 | −0.009 | 0.552 | 0.583 | −0.031 |

## 5 RESULTS

We evaluate PHIER on the three datasets and compare against 10 state-of-the art models, with our evaluation metric as binary state classification accuracy. Our experiments show that PHIER's learned predicate hierarchy leads to significantly improved performance, especially in the challenging settings of few-shot, out-of-distribution generalization as well as zero- and few-shot, real-world transfer.

### 5.1 IMPLEMENTATION

**PHIER.** For our model, we use the CLIP image encoder, CLIP text encoder, and BERT text encoder as our image, object text, and predicate text encoders, respectively. Our hyperbolic encoder consists of two hyperbolic linear layers with output dimensions of 256 and 128, and the final small MLP is a single layer. We use $\alpha = 0.05$ as our triplet loss coefficient, $\lambda = 10.0$ as our triplet loss margin, $\beta = 1.0$ as our regularization loss coefficient, and $\gamma = 0.1$ as our regularization margin. We train all models for 50 epochs using the AdamW optimizer with a learning rate of $1e^{-4}$ using a gradual warmup scheduler and cosine annealing decay. For the few-shot setting, we provide 5 examples of each novel predicate and train for 20 epochs with the same optimizer and learning rate.

**Baselines.** We use 10 state-of-the-art methods as baselines, ranging from supervised methods to pretrained large vision language models (VLMs). Of the supervised methods, RelViT (Ma et al., 2022) and SORNet (Yuan et al., 2022) are recent methods designed with a focus on state classification, while Re-Attention (Guo et al., 2020), Coarse-to-Fine (Nguyen et al., 2022), BUTD (Anderson et al., 2018), finetuned CLIP (Shen et al., 2021), and FiLM (Perez et al., 2018) are top-performing general VQA methods. The supervised models are all trained on the same data as PHIER, while the pretrained large VLMs, BLIP-2 (Li et al., 2023b), GPT-4V(OpenAI, 2023), and ViperGPT (Surís et al., 2023), are run inference-only. All methods are evaluated on both in-distribution and out-of-distribution queries, except for SORNet as its architecture does not allow classification of unseen states. Additional details on our baselines are provided in Appendix B.

### 5.2 COMPARISON TO PRIOR WORK

**Few-shot generalization accuracy.** In Table 1, we show comparisons of PHIER and prior work on CALVIN and BEHAVIOR datasets, in the 5-shot generalization setting. We split prior works into trained supervised methods (T) and inference-only pretrained VLMs (I). While PHIER yields comparable performance to top-performing prior works on the in-distribution test set, PHIER significantly outperforms all methods on the out-of-distribution test set. PHIER demonstrates a 22.5 percent

Table 2: Detailed breakdown of CALVIN and BEHAVIOR out-of-distribution results with accuracy on unseen object-predicates combinations and novel predicates.

| | CALVIN | | | BEHAVIOR | | |
|---|---|---|---|---|---|---|
| | All | Unseen Comb. | Novel Pred. | All | Unseen Comb. | Novel Pred. |
| PHIER (Ours) | **0.899** | **0.922** | **0.863** | **0.820** | **0.831** | **0.807** |
| Re-Attention | 0.674 | 0.777 | 0.612 | 0.652 | 0.718 | 0.581 |
| CoarseFine | 0.624 | 0.665 | 0.557 | 0.636 | 0.670 | 0.600 |
| BUTD | 0.585 | 0.622 | 0.563 | 0.712 | 0.766 | 0.653 |
| RelViT | 0.563 | 0.591 | 0.515 | 0.737 | 0.793 | 0.676 |
| CLIP | 0.546 | 0.596 | 0.463 | 0.632 | 0.684 | 0.576 |
| FiLM | 0.489 | 0.538 | 0.406 | 0.583 | 0.652 | 0.508 |

point improvement compared to the top-performing prior work in the out-of-distribution CALVIN setting, and a 8.3 percent point improvement in the out-of-distribution BEHAVIOR setting. Notably, PHIER sees a low drop in accuracy between in-distribution and out-of-distribution test sets, which we hypothesize is due to PHIER's structured representation space enabling generalization. We report more detailed results in Table 2, specifically the out-of-distribution accuracy of few-shot unseen object-predicate pairs and novel predicates. We see that for both categories of generalization, PHIER significantly outperforms prior works. On CALVIN, PHIER improves upon the top-performing prior work by 25.1 percent points in few-shot generalization to novel predicates. On BEHAVIOR, we see a 13.1 percent point improvement.

**Real world zero- and few-shot transfer accuracy.** In Table 3, we report the zero- and few-shot, real-world transfer results of models trained on the simulated BEHAVIOR dataset, and tested on the BEHAVIOR Vision Suite (Ge et al., 2024), a complex real-world benchmark. We compare PHIER with previous supervised methods, which have seen the same amount of train data, and find that PHIER significantly outperforms prior works on this challenging sim-to-real task across both zero- and few-shot settings. We conjecture that this is because PHIER learns more robust features for images—only features core to the specified state classification task are captured, and hence enables PHIER to generalize and remain invariant to the visual details in the real world. We additionally include results from pre-trained models, though we note that these models are our upper bound, as they are trained on large-scale real-world datasets with vast amounts of diverse data. Hence, these models inherently do not differentiate between in-distribution and generalization scenarios, as their training data overlaps significantly with both. We see that PHIER outperforms ViperGPT and BLIP-2 by 6.0% and 1.4% respectively on PHIER's zero-shot setting, showing the potential for a small model trained on significantly less data, to reach the performance level of large pre-trained models. However, GPT-4v outperforms PHIER by 10.4%. which we hypothesize is due to its model size and dataset scale. In the few-shot setting using only two examples, PHIER's performance improves significantly and narrows the gap with GPT-4v to just 0.9%. This further demonstrates that PHIER's inferred predicate hierarchy enables it to generalize efficiently to novel queries.

### 5.3 ABLATIONS

We ablate the components of PHIER in Table 4. Specifically, we begin by reporting the out-of-distribution results of a supervised model for state classification. We then test variants of PHIER, progressively adding each component: our object-centric encoder, predicate triplet loss, norm regularization, and finally the full model with the hyperbolic distance metric.

We see that each component encourages a structured and semantically relevant latent space for out-of-distribution generalization. The object-centric encoder localizes relevant objects in the scene, improving performance by 15.4 and 13.7 percent points on CALVIN and BEHAVIOR, respectively. Adding the predicate triplet loss helps PHIER encode pairwise relationships between predicates, improving performance by 13.1 and 8.2 percent point on CALVIN and BEHAVIOR. The addition of the norm regularization loss introduces hierarchical structure into the latent space, further improving performance by 6.5 and 4.7 percent points. Finally, we highlight the full PHIER equipped with the hyperbolic metric, which further enforces a tree-like hierarchy to emerge in latent space and yields the strongest generalization performance.

Table 3: We present zero- and few-shot generalization results on a real-world test set, when trained only on the BEHAVIOR dataset. PHIER outperforms all prior supervised models.

| | Zero-shot | | | Few-shot | | |
|---|---|---|---|---|---|---|
| | All | Unseen Comb. | Novel Pred. | All | Unseen Comb. | Novel Pred. |
| PHIER (Ours) | **0.608** | **0.632** | **0.585** | **0.703** | **0.714** | **0.691** |
| Re-Attention | 0.377 | 0.415 | 0.341 | 0.413 | 0.458 | 0.368 |
| CoarseFine | 0.490 | 0.485 | 0.494 | 0.553 | 0.562 | 0.543 |
| BUTD | 0.418 | 0.427 | 0.409 | 0.456 | 0.464 | 0.448 |
| RelViT | 0.556 | 0.579 | 0.528 | 0.603 | 0.654 | 0.552 |
| CLIP | 0.516 | 0.544 | 0.489 | 0.571 | 0.674 | 0.468 |
| FiLM | 0.459 | 0.480 | 0.438 | 0.513 | 0.542 | 0.484 |
| GPT-4V | 0.712 | 0.737 | 0.688 | —— | —— | —— |
| BLIP-2 | 0.594 | 0.591 | 0.597 | —— | —— | —— |
| ViperGPT | 0.548 | 0.538 | 0.557 | —— | —— | —— |

Table 4: Ablations of each component of PHIER and its effect on few-shot generalization.

| | CALVIN OOD | BEHAVIOR OOD |
|---|---|---|
| Supervised model | 0.473 | 0.516 |
| + Object-centric encoder | 0.627 | 0.653 |
| + Predicate triplet loss | 0.758 | 0.735 |
| + Norm regularization loss | 0.823 | 0.782 |
| + Hyperbolic metric (PHIER) | **0.899** | **0.830** |

## 5.4 DISCUSSION

We propose PHIER as a framework for incorporating predicate hierarchies into the latent space of state classification models. One important design decision is how explicitly the hierarchy should be enforced—PHIER softly encourages this structure with self-supervised losses, but does not impose hard constraints on the model's forward pass. We designed PHIER in this way, in order to allow hierarchical structure to emerge in the hyperbolic latent space based on the data, and capture more nuanced structure than a strict, discrete predicate hierarchy could. Quantitatively, we see that PHIER retains the ability to perform well on generalization tests, while qualitatively, we can visualize PHIER's learned predicate hierarchy in the latent space.

We note that PHIER is potentially limited by the accuracy of the language model in determining pairwise predicate relations. In this paper, we assume that language itself can differentiate the relationship between predicates, but there may be cases where visual cues from data also matter. Empirically, on the datasets we tested, we see that the LLM's predictions match our expectations of what the predicate hierarchy should be. Additionally, as PHIER's enforcement of this hierarchy is not explicit, it is still possible for PHIER to learn from data when the language model is incorrect. As a future direction, exploring environments where the dataset for state classification yields a unique predicate hierarchy, which we can encode through explicit enforcement in the model's forward pass, would showcase the effect of an explicitly hierarchical version of PHIER in generalization. In addition, exploring ways of training a model to infer the pairwise predicate relations with weak supervision, instead of injecting relation priors through a language model, could potentially give rise to a fully emergent and discovered predicate hierarchy.

## 6 CONCLUSION

PHIER tackles the challenge of few-shot out-of-distribution state classification by encoding predicate hierarchies into its latent space. Our proposed model, PHIER, learns language-informed image-predicate representations to generalize to novel predicates with few examples. Our experiments on CALVIN, BEHAVIOR, and a real-world test set demonstrate that PHIER significantly improves upon existing methods, particularly in highly difficult generalization cases. We show that using predicate hierarchies is a promising approach to enable more robust and adaptable state classification.

ACKNOWLEDGMENTS

We thank Weiyu Liu for providing valuable feedback. This work is in part supported by ONR N00014-23-1-2355, ONR YIP N00014-24-1-2117, ONR MURI N00014-22-1-2740, ONR MURI N00014-24-1-2748, NSF RI #2211258, and AFOSR YIP FA9550-23-1-0127. JH is also supported by the Knight Hennessy Scholarship and the NSF Graduate Research Fellowship.

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

# Supplementary for:
# Predicate Hierarchies Improve Few-Shot State Classification

The appendix is organized as the following. In Appendix A, we include additional PHIER results, details, and discussion. In Appendix B, we describe implementation of baseline methods, including supervised models, pretrained large vision language models, and ablation variants. In Appendix C, we present preliminaries on hyperbolic geometry. In Appendix D, we detail prompts used to extract knowledge of predicates from LLMs. Finally, in Appendix E, we list all states in our datasets, and show examples from the BEHAVIOR Vision Suite Ge et al. (2024).

## A  PHIER Results and Details

### A.1  Model Details

PHIER's image and text encoders are initialized with pretrained CLIP (Radford et al., 2021) and BERT (Devlin, 2018) weights, respectively. The hyperbolic linear layers are initialized following the approach of Shimizu et al. (2020), with the weights drawn from a normal distribution centered at zero with a standard deviation $(2nm)^{-\frac{1}{2}}$, where $m$ and $n$ are the input and output sizes of the layer, and the biases set to the zero vector. The linear layer in the small MLP is initialized by the standard Kaiming initialization. All of the parameters in PHIER are trainable and updated during training.

PHIER disentangles the conditioning of the image on the full state classification query into two distinct ones: one that identifies the relevant objects and another that focuses on key features for the given predicate. While we use MaskCLIP to identify the relevant entities, PHIER's contribution lies in the decomposition of the query into object and predicate components, enabling it to faithfully identify the relevant entities and extract features based on the predicate.

### A.2  Comparison on Manually Collected Real-world Dataset

We collect a small real-world dataset with 100 examples, consisting of 4 examples for each of the out-of-distribution BEHAVIOR states, to test our method's ability to perform zero-shot real-world transfer after training on simulated datasets alone. See Figure 4 for examples. In Table **??**, we observe similar trends as in the BEHAVIOR Vision Suite evaluation in the main text, even with a simpler dataset. PHIER significantly outperforms prior supervised baselines. However, as expected, pre-trained models trained on large-scale real-world data outperform PHIER.

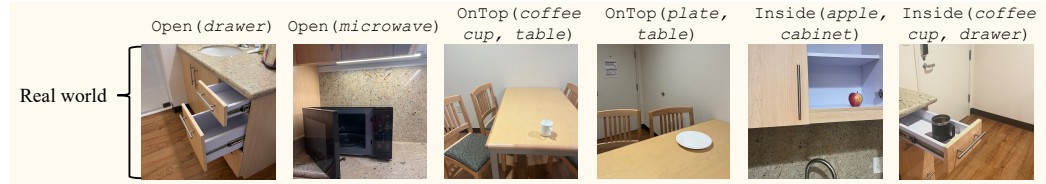

Figure 4: Examples from our manually collected real-world dataset.

### A.3  Ablation Study on Example Count

We study the effect of varying the number of examples used in the few-shot setting. We added new ablation experiments with 0, 1, 2, 3, 4, 5, and 10-shot generalization performance on both CALVIN and BEHAVIOR environments. The results in Figure 5 show that PHIER consistently outperforms prior works across all numbers of examples. Notably, in the CALVIN environment, PHIER's performance plateaus as the number of examples increases, indicating that the method requires only a few examples to adapt effectively to unseen scenarios.

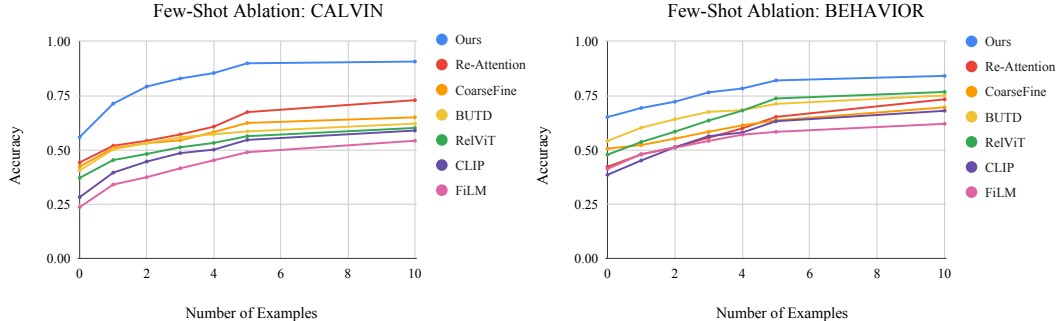

Figure 5: Ablations varying number of examples given in few-shot setting for CALVIN and BEHAV-IOR environments.

## A.4 ABLATION STUDY WITH REMOVED COMPONENTS

We add an ablation study that evaluates the impact of removing individual components of PHIER to evaluate their contributions. We compare PHIER with four variants, (1) without the object-centric encoder, (2) without the hyperbolic latent space, (3) without the norm regularization loss, and (4) without the predicate triplet loss. We report results in Table 5. We see that without our object-centric design, performance drops significantly in both ID and OOD settings, emphasizing the importance of object-centric encoders for improved representation and reasoning. In addition, we show that removing each of the self-supervised losses leads to much weaker generalization capability. Finally, we observe reduced generalization performance without PHIER 's hyperbolic latent space and hyperbolic norm regularization loss, demonstrating that the hyperbolic space facilitates better handling of hierarchical relationships. These results validate that each component contributes meaningfully to PHIER's performance, particularly in improving OOD generalization.

Table 5: Ablations of each component of PHIER and its effect on few-shot generalization.

|  | CALVIN | | | BEHAVIOR | | |
|---|---|---|---|---|---|---|
|  | ID ↑ | OOD ↑ | ID-OOD ↓ | ID ↑ | OOD ↑ | ID-OOD ↓ |
| PHIER (Ours) | 0.945 | 0.899 | 0.046 | 0.859 | 0.820 | 0.039 |
| - Object-centric encoder | 0.786 | 0.704 | 0.082 | 0.703 | 0.659 | 0.044 |
| - Predicate triplet loss | 0.867 | 0.601 | 0.266 | 0.774 | 0.624 | 0.150 |
| - Norm regularization loss | 0.914 | 0.823 | 0.091 | 0.834 | 0.782 | 0.052 |
| - Hyperbolic metric | 0.903 | 0.784 | 0.119 | 0.803 | 0.761 | 0.042 |

## A.5 FEW-SHOT GENERALIZATION TO NOVEL OBJECTS

We expand our CALVIN and BEHAVIOR experiments to evaluate accuracy on few-shot generalization on novel objects in Table 6. The queries with these novel objects are listed in Table 7. As in our experiments on unseen combinations and novel predicates, we observe that PHIER significantly outperforms prior baselines on unseen objects. Specifically, PHIER improves upon the top-performing prior work by 21.8 percent points on CALVIN and 13.5 percent point on BEHAVIOR. These results demonstrate that PHIER improves generalization to both novel objects and predicates, further highlighting the benefit of our object-centric encoder and inferred predicate hierarchy.

## A.6 VISUALIZATIONS ON THE INFERRED PREDICATE HIERARCHY

In Figure 6, we visualize the joint image-predicate space for BEHAVIOR on the Poincaré disk, highlighting the hierarchical semantic structure captured by PHIER's embeddings. By grouping the joint image-predicate embeddings by predicate, we uncover the inferred predicate hierarchy. For

Table 6: We present novel object generalization results of PHIER and prior works on CALVIN and BEHAVIOR environments.

|  | CALVIN | BEHAVIOR |
|---|---|---|
| PHIER (Ours) | **0.851** | **0.781** |
| Re-Attention | 0.633 | 0.608 |
| CoarseFine | 0.562 | 0.632 |
| BUTD | 0.584 | 0.646 |
| RelViT | 0.497 | 0.642 |
| CLIP | 0.506 | 0.595 |
| FiLM | 0.411 | 0.521 |

Table 7: All states with novel objects (bolded) in the CALVIN and BEHAVIOR datasets.

| Dataset | Predicate | Object 1 | Object 2 |
|---|---|---|---|
| CALVIN | OnTop | **red block** | table |
| | Stacked | **red block** | blue block |
| | Stacked | **red block** | pink block |
| | TurnedOn | **led** | – |
| BEHAVIOR | Inside | **box** | bottom cabinet |
| | Inside | **can** | bottom cabinet |
| | OnTop | **bottle** | breakfast table |
| | OnTop | **bottle** | **chair** |
| | OnTop | **box** | breakfast table |
| | OnTop | **bread** | breakfast table |
| | OnTop | **can** | **chair** |
| | Open | **refrigerator** | – |

instance, we see that embeddings for `NextTo` are positioned closer to the origin compared to those for `OnLeft`, accurately reflecting their hierarchical relationship——`OnLeft` is a more specific case of `NextTo`. Furthermore, embeddings for `Touching` are nearest to the origin, consistent with its role as the most general predicate. For example, when one object is `Inside` or `OnTop` of another, they are inherently `Touching`. Similarly, objects that are `NextTo` or `OnLeft` are also frequently `Touching`. This visualization demonstrates that PHIER captures not only semantic structure but also nuanced hierarchical relationships between predicates.

We further analyze the embeddings for novel predicates after few-shot learning with only five examples. Notably, even with such limited data, PHIER successfully integrates these novel predicates into the latent space and aligns them with their learned counterparts in semantically consistent regions (e.g., `OnRight` is near `OnLeft`). By aligning these predicates in similar regions, PHIER is able to leverage its existing knowledge of relevant features for learned predicates (e.g., `OnLeft`) to reason about novel predicates (e.g., `OnRight`). This alignment highlights that PHIER effectively encodes the relationships between pairwise predicates in the latent space, enabling generalization to novel predicates with minimal examples.

## A.7 IN-DISTRIBUTION PERFORMANCE

Here, we discuss the in-distribution performance of PHIER in Table 1 of the main text. We note that in the in-distribution (ID) setting of CALVIN, PHIER outperforms all prior works except Re-Attention, with only a small margin of $1.4\%$. In the out-of-distribution (OOD) setting, which is our primary focus, PHIER outperforms Re-Attention by a significant $22.5\%$. Similarly, on ID BEHAVIOR, PHIER performs comparably to top-performing prior works, surpassing all except RelViT by $0.7\%$; however, in the OOD setting we focus on, PHIER outperforms RelViT by $8.3\%$. We highlight that PHIER performs comparably to top-performing prior works in the ID setting, while significantly improving the OOD performance. We focus on the few-shot generalization task and

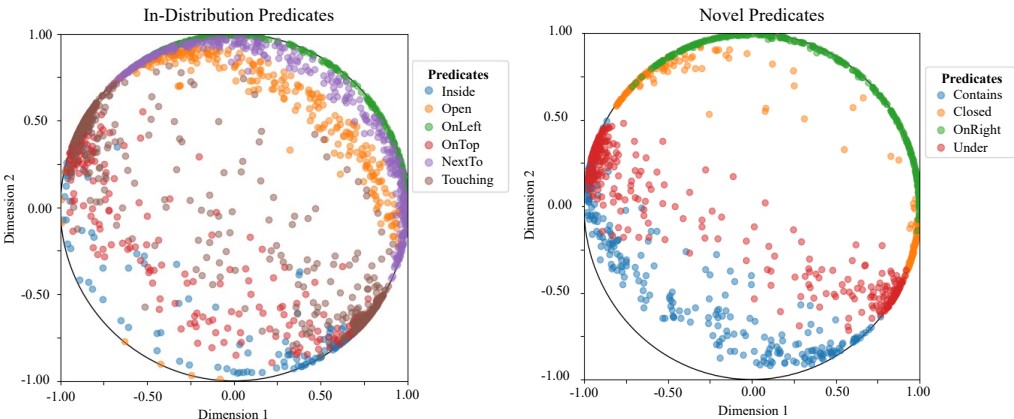

Figure 6: Visualizations of the joint image-predicate space for BEHAVIOR on the Poincaré disk, revealing that PHIER learns a meaningful predicate hierarchy. The novel predicate embeddings are visualized after few-shot learning with 5 examples.

design our method to enforce bottlenecked representations (via a joint image-predicate space), while acknowledging that this might include tradeoffs on ID performance to avoid overfitting to the train distribution.

We also analyze specific cases where PHIER underperforms on ID examples. For instance, in CALVIN, we hypothesize that PHIER may struggle with tasks that the baselines may memorize due to their less constrained representations. We show an example in Figure 7, and note that for the ID query, `TurnedOn(lightbulb)`, Re-Attention correctly predicts `True`, while PHIER predicts `False`. However, for the out-of-distribution query, `TurnedOff(lightbulb)`, which is linguistically similar but semantically opposite, PHIER generalizes successfully while Re-Attention struggles to adapt. We conjecture that Re-Attention may predict that `TurnedOn(lightbulb)` is `True` based solely on the existence of the bulb at the location, instead of learning that the state of the lightbulb depends on its color (yellow is on and white is off). In contrast, we see that although PHIER's constrained representation may slightly limit learning capacity for ID settings, PHIER has the potential to conduct better compositional reasoning in OOD scenarios, where PHIER significantly outperforms baselines.

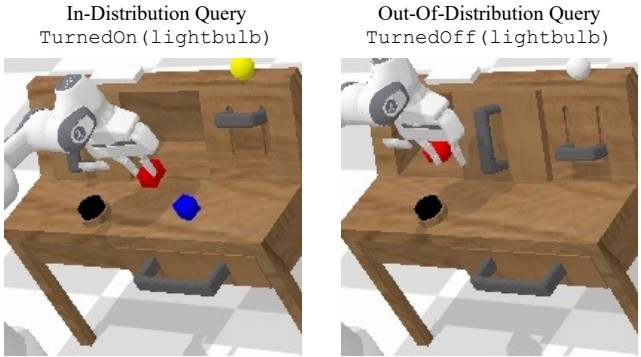

Figure 7: An example of the ID query, `TurnedOn(lightbulb)`, and OOD query with a novel predicate, `TurnedOff(lightbulb)`.

## A.8  OBJECT-CENTRIC ENCODER PERFORMANCE

We see empirically that PHIER's object-centric encoder performs well even on environments with significant distribution shifts, such as CALVIN. In Figure 8, we show an example of how the encoder localizes objects in CALVIN. To adapt to environments with even larger distribution shifts where the

performance may decrease, we note that PHIER's object-centric encoder can be finetuned with more data as well.

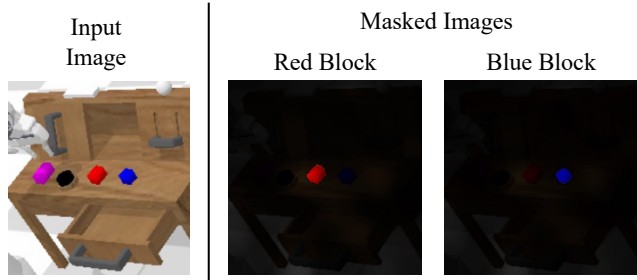

Figure 8: PHIER's object-centric encoder in the CALVIN environment.

## B    BASELINE DETAILS

For all of our baseline methods, we preprocess our input queries by converting the states into questions using the following templates:

- For unary states: "Is the {*object*} {predicate}"
- For binary states: "Is the {*object 1*} {predicate} the {*object 2*}"

### B.1    SUPERVISED METHODS

We train all of the supervised baselines on the same training data as our method. Below, we describe each baseline and provide implementation details:

**BUTD (Anderson et al., 2018).** BUTD uses bottom-up attention to extract image features for important image regions and then top-down attention to focus on image regions based on the input query. We follow the original method, using Faster R-CNN pretrained on Visual Genome to extract bottom-up features for the top 36 image regions. For the text features, we embed the preprocessed text queries using 300-dimension word embeddings, initialized with pretrained GloVe vectors, and a GRU. The image and text features are then fed into model, based on the PyTorch implementation of the BUTD model for VQA (Yu et al., 2020)[*].

**CLIP (Shen et al., 2021).** We use pretrained CLIP vision and text encoders to extract features for the input image and query, respectively. These features are concatenated and passed through a small 2-layer network with a hidden layer of dimension 256 for state classification.

**CoarseFine (Nguyen et al., 2022).** Coarse to Fine learns to reason about scenes with complex semantic information by extracting image and text features at multiple levels of granularity. We follow the official implementation of the Coarse to Fine reasoning framework and use Faster R-CNN to extract image-level features and GRU with 300-dimensional GloVe embeddings to extract question-level features, which are then fed into the model[†].

**FiLM (Perez et al., 2018).** FiLM conditions an input image on text by applying learned transformations to the image features. We use a pretrained ViT-16 image encoder and BERT text encoder to extract image and query features. Then, a FiLM layer is applied to condition the image features on the query features, and the conditioned features are passed through a small 2-layer network with a hidden layer of dimension 256 for final prediction.

**Re-Attention (Guo et al., 2020).** Re-Attention introduces an attention mechanism to re-attend to objects in the images, based on the answer to the question. We follow the original implementation by using a Faster R-CNN model pretrained on the Visual Genome dataset to extract object-level image features, and 512-dimensional LSTM initialized with 300-dimensional GloVe embeddings to extract query features [‡].

**RelViT (Ma et al., 2022).** RelViT enhances the reasoning ability of vision transformers by introducing a concept-feature dictionary that enables efficient image feature retrieval during training. This supports a global task to promote relational reasoning and a local task to learn semantic object-centric correspondences. We use the official implementation, with Faster R-CNN to extract image region features, MCAN-Small as our VQA model, and the ImageNet1K-pretrained PVTv2b2 as our vision backbone [§].

**SORNet (Yuan et al., 2022).** SORNet extracts object-centric representations from input RGB images, conditioned on a set of object queries represented as images of the objects, to enable generalization to unseen objects on various spatial reasoning tasks. It performs state classification by training readout networks to predict spatial relations based on the object embeddings. For a fair comparison to our method and other baselines, we use MDETR (Kamath et al., 2021) to detect regions corresponding to object text, resize then to $32 \times 32$, and then use them as the input object images to train SORNet.

---

[*]https://github.com/MILVLG/bottom-up-attention.pytorch
[†]https://github.com/aioz-ai/CFR_VQA
[‡]https://github.com/gwy-nk/Re-Attention
[§]https://github.com/NVlabs/RelViT

We train readout networks for each training state in our dataset [¶]. Since SORNet requires training a separate network for each predicate, we only evaluate it on in-distribution states.

## B.2 PRETRAINED LARGE VISION LANGUAGE MODELS (VLM)

All of the pretrained large VLM baselines are evaluated inference-only.

**BLIP-2 (Li et al., 2023b).** We use BLIP-2 leveraging the OPT-2.7b language model and treat VQA as an open-ended answer generation problem. The input image is provided along with a query using the following format: "Question: {state query as a question} Answer:"

**GPT-4V (OpenAI, 2023).** We provide GPT-4V with the input image and a prompt based on the following template:

---

**Prompt Template For GPT-4V Inference**

```
Given an image of a scene, you will answer a question regarding the states
and relationships of objects in the scene.  The question is the following:

{state query as a question}

You need to carefully examine the image, thoughtfully consider the objects
in the scene, and analyze their states and relationships before answering
the question.

Provide your answer as True or False, and strictly follow this response
format:
Answer:  [insert your answer as True or False here]
Reasoning:  [insert your reasoning here]
```

---

Figure 9: Prompt template for GPT-4V experiments.

**ViperGPT (Surís et al., 2023).** We use the official ViperGPT implementation with Blip-2 Flan-T5 XXL as the pretrained model and GPT-4 for code generation. Our data is formatted according to the ViperGPT specifications, with the input image and query as a question.

## B.3 ABLATION DETAILS

Here, we provide a clear breakdown of our ablation model architectures from Table 4 and explain how we add each component.

**Supervised model. We start with a supervised baseline model, which uses an image encoder and text incoder initialized with CLIP and BERT weights, respectively.The embeddings from both encoders are concatenated and passed through a small MLP with three linear layers for classification, and the full model is trained with a binary cross-entropy loss based on the ground truth labels (True or False). We then progressively add each component of PHIER.**

**+ Object-centric encoder.** First, we incorporate the object-centric encoder by replacing the image encoder, text encoder, and concatenation step with our proposed object-centric encoder, while retaining the MLP and loss.

**+ Predicate triplet loss.** Next, we introduce the predicate triplet loss by adding this term to the total loss function without changing the architecture.

**+ Norm regularization loss.** We further add the norm regularization loss to get the total loss function with all components, as described in Section 3

**+ Hyperbolic metric.** Finally, we lift the scene representation to hyperbolic space using an exponential map and replace the first two linear layers in the MLP with two hyperbolic linear layers of the same size. We also use the Poincaré distance metric instead of the Euclidean metric in the self-supervised losses, yielding our final model (PHIER).

---

[¶]https://github.com/wentaoyuan/sornet

## C  HYPERBOLIC GEOMETRY PRELIMINARY

We briefly introduce the Poincaré ball model of hyperbolic space and hyperbolic neural networks. For a more detailed explanation, we refer the reader to Cannon et al. (1997) and Ganea et al. (2018).

As discussed in the main text, the Poincaré ball is a $d$-dimensional ball of radius 1, $\mathbb{P}^d = \{x \in \mathbb{R}^n : ||x|| < 1\}$, where $|| \cdot ||$ is the Euclidean norm. The ball is equipped with the metric tensor $g_{\mathbf{P}} = (\lambda_x)^2 g_{\mathbf{e}}$, where $\lambda_x = \frac{2}{1-||x||^2}$ is the conformal factor and $g_e$ is the Euclidean metric tensor (i.e., the Euclidean dot product). This induces the Poincaré distance $\mathbf{d_P}$ between two points $x, y \in \mathbb{P}^d$ as follows:

$$\mathbf{d_P}(x, y) = \cosh^{-1}\left(1 + 2\frac{||x - y||^2}{(1 - ||x||^2)(1 - ||y||^2)}\right)$$

**Möbius addition.** On the Poincaré ball, Euclidean operations such as addition and multiplication have equivalents to ensure that all operations remain within the hyperbolic space and respect its geometry. Instead of using standard Euclidean addition, Möbius addition is used, which ensures that the sum of two points on the Poincaré ball still lies within the ball. The Möbius addition for any two points $x, y \in \mathbb{P}^d$ is defined as:

$$x \oplus y := \frac{(1 + 2\langle x, y\rangle + ||y||^2)x + (1 - ||x||)^2 y}{1 + 2\langle x, y\rangle + ||x||^2||y||^2}$$

**Exponential and logarithmic maps.** To perform operations in hyperbolic space, we use exponential and logarithmic maps to map Euclidean vectors to the hyperbolic space, and vice versa. For any point $z \in \mathbb{P}^d$, the closed form expression of the exponential and logarithmic maps centered around $z$ are defined as:

$$\exp_z(y) = z \oplus \left(\tanh\left(\frac{\lambda_z ||v||}{2}\right)\frac{v}{||v||}\right)$$

$$\log_z(y) = \frac{2}{\lambda_z}\tanh^{-1}(|| -z \oplus y||)\frac{-z \oplus y}{|| -z \oplus y||}$$

In practice, we use the maps centered at 0, $\exp_{\mathbf{0}}$ and $\log_{\mathbf{0}}$, to transition between Euclidean space and the Poincaré ball.

**Hyperbolic neural networks.** Ganea et al. (2018) proposes hyperbolic neural networks by defining hyperbolic equivalents of linear maps and bias translations. The hyperbolic linear map $M^{\otimes} : \mathbb{R}^n \to \mathbb{R}^m$ of any point $x \in \mathbb{P}^d$ on the Poincaré ball is defined as:

$$M^{\otimes}(x) = (1/\sqrt{c})\tanh\left(\frac{||Mx||}{||x||}\tanh^{-1}\left(\sqrt{c}||x||\right)\right)\frac{Mx}{||Mx||}$$

The translation of a point $x \in \mathbb{P}^d$ by a bias $b \in \mathbb{P}^d$ as:

$$x \oplus b = \exp_x\left(\frac{\lambda_{\mathbf{0}}}{\lambda_x}\log_{\mathbf{0}}(c)\right)$$

The hyperbolic linear layer is then defined as $M^{\otimes}(x) \oplus b$. To build a hyperbolic neural network, one simply has to map representations to the Poincaré ball using $\exp_{\mathbf{0}}$, apply hyperbolic linear layers, and then map back to Euclidean space using $\log_{\mathbf{0}}$.

**Disk area of hyperbolic space.**

We provide further details on why the exponential growth of the disc area in hyperbolic space provides a natural and efficient way to represent trees. Note that for a regular tree with a constant branching factor $b$, the number of nodes increases exponentially with the distance from the root, as $(b + 1)b^{\updownarrow-1}$. We can embed trees in hyperbolic space, as they mirror this exponential growth. For instance, in a two-dimensional hyperbolic space with constant curvature $K = -1$, the circumference of a disc with radiuds $r$ is $2\pi \sinh r$ while the area of a disc is $2\pi(\cosh r - 1)$. Since $\sinh r = \frac{1}{2}(e^r - e^{-r})$ and $cosh r = \frac{1}{2}(e^r + e^{-r})$, both the circumference and area of the disc grow exponentially with the radius.

This exponential growth allows us to efficiently embed tree structures in hyperbolic space: nodes that are $\updownarrow$ levels from the root can be placed on the hyperbolic disc with a radius proportional to its level

↕, while nodes less than ↕ levels within the sphere. Thus, we see how this property allows hyperbolic space to serve as a continuous representation of discrete trees.

**Connection between hyperbolic space and hierarchical structure.** We highlight several prominent prior works who have made theoretical connections between hyperbolic space and trees. Mathematical works such as Gromov (1987), Dyubina & Polterovich (2001), and Hamann (2018) prove that any finite tree can be embedded into a finite hyperbolic space with approximately preserved distances. A key property of hyperbolic space is its exponentially growing distance, and they show that this underlying property makes hyperbolic space well-suited to model hierarchical structures. Furthermore, works such as Sala et al. (2018) and Chami et al. (2020) propose concrete approaches to embed any tree in hyperbolic space with arbitrarily low distortion, establishing upper upper and lower bounds for distortion and further demonstrating the effectiveness of hyperbolic space for hierarchical modeling.

Notably, Nickel & Kiela (2017) were among the first to explore learning hierarchical representations in hyperbolic space. They found that for data with latent hierarchies, embeddings on the Poincaré ball outperform Euclidean embeddings significantly in terms of representation capacity and generalization ability. Since then, hyperbolic spaces have been increasingly explored for modeling hierarchies across various domains, including NLP (Ganea et al., 2018; Nickel & Kiela, 2018; Tifrea et al., 2018) and computer vision (Khrulkov et al., 2020; Ermolov et al., 2022), with substantial empirical evidence supporting its efficiency and suitability for modeling hierarchical structures in comparison to Euclidean space. We believe that these prior works provide strong theoretical justification and empirical support for the connection between hyperbolic space and hierarchical structure, which inspires our method.

**Implementation.** We implement our hyperbolic encoder using the Geoopt package (Kochurov et al., 2020), which provides functions and optimization methods for hyperbolic space [∥].

---

[∥]https://github.com/geoopt/geoopt

## D  LLM PROMPTS FOR SELF-SUPERVISED LOSSES

Here, we present the prompt templates used to extract explicit knowledge of predicates from LLMs. In Figure 10, we describe the prompt used to determine the assignment (anchor, positive predicate, and negative negative) for a given triplet of predicates, used in the predicate triplet loss. In Figure 11, we show the prompt used to determine the hierarchy among a predicate triplet based on specificity, for the norm regularization loss. We query the LLM once before training starts to retrieve the predicate triplet pairs and hierarchy, hence training is not affected by LLM queries.

---

**Prompt Template For Predicate Triplet Assignment**

```
You are given an anchor text query that describes a state of a scene.  Given
two other text queries describing the state of a scene, you will help
determine which of the two queries is more similar to the anchor query.

Consider the semantic meaning of the states and the specific aspects of
the scene they describe.  Additionally, think about how many objects and
what kinds of object properties and features you would need to verify if
evaluating these states against an image.

The anchor query is the following:  {anchor}

The other two queries are:
Query 1:  {query1}
Query 2:  {query2}

You must choose one of the queries as your answer.  Respond using the
following format:
Answer:  [Query 1 or Query 2]
```

---

Figure 10: Prompt template for inferring the predicate relations among a triplet with GPT-4.

---

**Prompt Template For Triplet Hierarchy Ranking**

```
You are an expert in scene understanding and state hierarchy determination.
Given three text descriptions each outlining a potential state of a
scene, your task is to establish a hierarchy among these descriptions by
identifying which one is the most general, which is the most specific, and
which lies in between.

Consider the following when determining the hierarchy:
- The variety and number of objects required by the state.
- The important features of the objects and/or relationships between the
objects.
- The level of detail provided about the scene.
- The semantic meaning of each description.

Your goal is to rank these descriptions in order of specificity, from least
specific (1) to most specific (3).

The three descriptions are:
1.  {anchor}
2.  {query1}
3.  {query2}

You must provide your ranking using the following format:
Least Specific:  [content of Description 1, 2, or 3]
Intermediate Specific:  [content of Description 1, 2, or 3]
Most Specific:  [content of Description 1, 2, or 3]
```

---

Figure 11: Prompt template for inferring the hierarchy among a triplet with GPT-4.

# E   DATASET DETAILS

## E.1   DATASET STATES

In Tables 8 and 9, we provide all of the states included in the CALVIN and BEHAVIOR datasets.

Table 8: All states included in the CALVIN dataset.

| State Type | Predicate | Object 1 | Object 2 |
|---|---|---|---|
| ID | Lifted | blue block | – |
| ID | OnRight | slider | – |
| ID | Open | drawer | – |
| ID | TurnedOn | lightbulb | – |
| ID | Inside | blue block | drawer |
| ID | Inside | pink block | drawer |
| ID | OnTop | blue block | table |
| ID | Stacked | blue block | pink block |
| ID | Stacked | blue block | red block |
| OOD | Closed | drawer | – |
| OOD | Lifted | pink block | – |
| OOD | OnLeft | slider | – |
| OOD | TurnedOff | lightbulb | – |
| OOD | Inside | blue block | slider |
| OOD | OnTop | pink block | table |
| OOD | Stacked | pink block | blue block |
| OOD | Under | table | blue block |

## E.2   BEHAVIOR VISION SUITE VISUALIZATIONS

We include additional examples from BEHAVIOR Vision Suite Ge et al. (2024) in Figure 12.

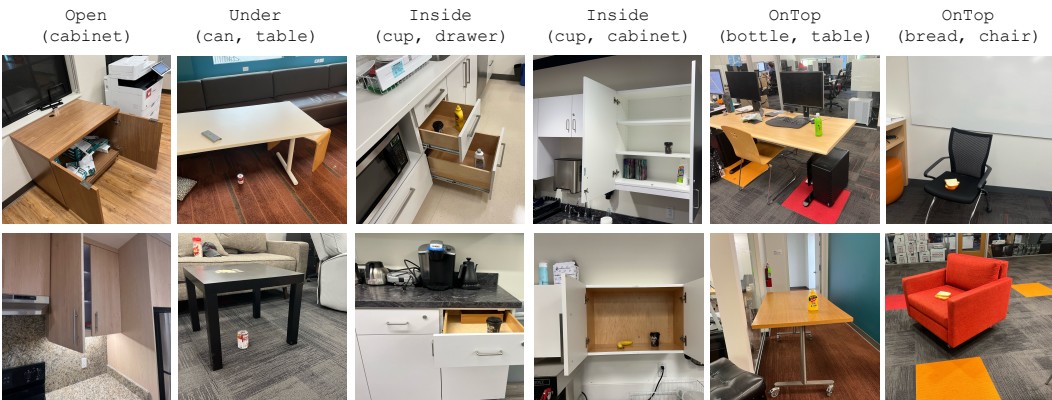

Figure 12: Visualizations of state classification tasks from the real-world BEHAVIOR Vision Suite dataset.

Table 9: All states included in the BEHAVIOR dataset.

| State Type | Predicate | Object 1 | Object 2 |
|---|---|---|---|
| ID | Open | bottom cabinet | – |
| ID | Open | drawer | – |
| ID | Open | microwave | – |
| ID | Open | oven | – |
| ID | Open | top cabinet | – |
| ID | Inside | apple | top cabinet |
| ID | Inside | club sandwich | microwave |
| ID | Inside | pizza | microwave |
| ID | Inside | plate | bottom cabinet |
| ID | Inside | plate | bottom cabinet no top |
| ID | NextTo | apple | coffee cup |
| ID | NextTo | coffee cup | cola bottle |
| ID | NextTo | croissant | cheesecake |
| ID | NextTo | pizza | microwave |
| ID | NextTo | plate | coffee cup |
| ID | OnLeft | apple | coffee cup |
| ID | OnLeft | coffee cup | cola bottle |
| ID | OnLeft | croissant | cheesecake |
| ID | OnLeft | pizza | microwave |
| ID | OnLeft | plate | coffee cup |
| ID | OnTop | apple | plate |
| ID | OnTop | cheesecake | plate |
| ID | OnTop | coffee cup | breakfast table |
| ID | OnTop | cola bottle | countertop |
| ID | OnTop | plate | breakfast table |
| ID | Touching | apple | plate |
| ID | Touching | cheesecake | plate |
| ID | Touching | coffee cup | breakfast table |
| ID | Touching | cola bottle | breakfast table |
| ID | Touching | croissant | plate |
| OOD | Closed | bottom cabinet | – |
| OOD | Closed | drawer | – |
| OOD | Closed | microwave | – |
| OOD | Closed | top cabinet | – |
| OOD | Contains | bottom cabinet | plate |
| OOD | Contains | drawer | plate |
| OOD | Contains | top cabinet | drawer |
| OOD | Inside | apple | microwave |
| OOD | Inside | coffee cup | top cabinet |
| OOD | Inside | plate | microwave |
| OOD | NextTo | apple | plate |
| OOD | NextTo | plate | microwave |
| OOD | OnTop | coffee cup | plate |
| OOD | OnTop | apple | breakfast table |
| OOD | OnTop | apple | microwave |
| OOD | OnLeft | apple | plate |
| OOD | OnLeft | coffee cup | apple |
| OOD | OnRight | apple | coffee cup |
| OOD | OnRight | coffee cup | cola bottle |
| OOD | OnRight | plate | coffee cup |
| OOD | Touching | apple | breakfast table |
| OOD | Touching | coffee cup | plate |
| OOD | Under | breakfast table | coffee cup |
| OOD | Under | breakfast table | plate |
| OOD | Under | plate | apple |

