# OpenReview forum: "Predicate Hierarchies Improve Few-Shot State Classification"
_ICLR.cc/2025/Conference — ICLR 2025 Poster_

### Official Review · Reviewer_V1Qj · 2024-10-31

**Soundness:** 3
**Presentation:** 4
**Contribution:** 3
**Rating:** 6
**Confidence:** 4

**Summary:**

This paper proposes PHIER, a method for few-shot state classification by learning hierarchical predicate representations in a hyperbolic latent space. The proposed method consists of three main modules:

 A) an object-centric scene encoder based on CLIP.

B) a triplet loss for predicate similarities based on rankings retrieved from an LLM.

C) A Latent space with hyperbolic structure for encoding predicate hierarchies.


Experiments on two robotic simulation environments and a new real-world image dataset are performed to conclude the superior performance of PHIER in contrast to other supervised models or VLMs, especially in a few-shot setting.

**Strengths:**

- The paper is very well written, and the method is clear and easy to follow.
- Most design choices are justified and effective, as shown with ablations.
- Good experimental setup and results in simulation. The results regarding novel predicates show that the learned latent space is efficient, especially compared to baselines.

**Weaknesses:**

- The ablation section needs more details
    - In its current state, is not clear what type of architecture the ablation baseline models utilize. What is the architecture of the supervised model? Just an image encoder and the predicate encoder followed by the supervised loss? This needs clarification. What is the hyperbolic linear layer replaced with? Simple MLPs?
- The network architecture is unclear from the current state of the method section. What are the trainable parameters? Where exactly do you introduce new parameters in the architecture? Only to project into the latent space and back? Including this in Figure 2, for instance, would make it easier for the reader to grasp this.
- Although the results in simulation are convincing, I have a few concerns:
    - The evaluation scenarios are all very simple and do not include any distractors.
    - You do not evaluate on unseen objects. The object-centric encoder should easily allow it to adapt to new unseen objects. Experiments in that regard could further strengthen the contribution of the paper.
    - The SORNet baseline uses an Mdetr object detector. Why do you not use your object-centric descriptor to generate the reference embeddings for SORNet? This would allow for a fairer comparison. This is a minor concern as the method does not require these representations in the first place.
    - You do not evaluate GPT4-V on the real-world benchmark. GPT4-V is trained on real-world data, presenting a strong baseline in this case.
    - It would be very interesting to see the few-shot transfer of the method to real-world tasks. You show that few-shot transfer works well in Simulation, so showing that it also works for sim to real transfer would strengthen the contribution.
    - The zero-shot results in real-world are not convincing. Although you outperform baselines by a large margin (which are frequently worse than random guessing), the success rate is still low. This is why I would recommend evaluating the method on few-shot transfer to real-world.

Overall, the simulation results are convincing, but for this paper to have a higher impact, more challenging evaluation settings and additional real-world experiments are required. Nevertheless, the results regarding few-shot transfer to new predicates are very promising.

**Questions:**

- From the text, it remains unclear to me if the object-centric encoder is part of the contribution or an application of Mask CLIP. Please clarify.
- An additional visualization/analysis of the learned Poincare Ball structure as part of the Appendix would highlight the core contribution of the method and further strengthen the paper.
- Why do you choose a Bert encoder over the CLIP text encoder?
- How sensitive is the model to hyperparameters (loss weights) in different envs? How did you determine them?
- How fast is the training when you have to query the LLM for hierarchy ranking during Training? Do you use some buffer to not reprompt the LLM every time?
- Why does the  GPT4-V reasoning happen after the model gives its answer?
- Is the ID evaluation done after few shot training or before?
- How does the object-centric encoder work for environments with significant distribution shifts, such as CALVIN?

---

> ### Author Response · Authors · 2024-11-21
> **Response to Reviewer V1Qj (pt 1)**
>
> **Q: Evaluation on a larger real-world dataset in zero-shot and few-shot settings.**
>
> A: Thank you for this suggestion! We have added additional evaluation on BEHAVIOR Vision Suite [1], a new and more complex real-world benchmark. The dataset consists of 500 examples with significantly more diverse scenes and distractor objects. Specifically, compared to our train data, this benchmark includes 10 unseen combinations (171 examples), 10 novel predicates (166 examples), and 10 novel objects (163 examples). See dataset examples in the updated Figure 3 in the main text as well as in Figure 12 in the Appendix.
> We present the zero- and few-shot real-world transfer results of PHIER and previous supervised models trained on the simulated BEHAVIOR dataset, and then tested on this real-world dataset. For a comprehensive comparison, we also evaluate pre-trained models as our upper bound.
>
> |              | Zero-Shot |        |            | 2-Shot |        |            |
> |--------------|-----------|--------|------------|--------|--------|------------|
> |              | All       | Unseen Comb. | Novel Pred. | All    | Unseen Comb. | Novel Pred. |
> | Ours         | 0.608     | 0.632  | 0.585      | 0.703  | 0.714  | 0.691      |
> | Re-Attention | 0.377     | 0.415  | 0.341      | 0.413  | 0.458  | 0.368      |
> | CoarseFine   | 0.490      | 0.485  | 0.494      | 0.553  | 0.562  | 0.543      |
> | BUTD         | 0.418     | 0.427  | 0.409      | 0.456  | 0.464  | 0.448      |
> | RelViT       | 0.553     | 0.579  | 0.528      | 0.603  | 0.654  | 0.552      |
> | CLIP         | 0.516     | 0.544  | 0.489      | 0.571  | 0.674  | 0.468      |
> | FiLM         | 0.459     | 0.480   | 0.438      | 0.513  | 0.542  | 0.484      |
> | GPT-4V       | 0.712     | 0.737  | 0.688      | --     | --     | --         |
> | BLIP-2       | 0.599     | 0.602  | 0.597      | --     | --     | --         |
> | ViperGPT     | 0.553     | 0.538  | 0.568      | --     | --     | --         |
>
> We observe similar trends as in our manually-collected real-world setup, with PHIER significantly outperforming prior supervised baselines on this challenging sim-to-real task across both zero- and few-shot settings. We conjecture that this is because PHIER learns more robust features for images—only features core to the specified state classification task are captured, and hence enables PHIER to generalize and remain invariant to the visual details in the real world. However, as expected, pre-trained models trained on large-scale real-world data outperform PHIER due to their large training data corpus. We added experiment results and examples of the dataset in Section 5 of the main text and Appendix Section A.2.
>
> [1] Ge, Yunhao, Yihe Tang, Jiashu Xu, Cem Gokmen, Chengshu Li, Wensi Ai, Benjamin Jose Martinez et al. "BEHAVIOR Vision Suite: Customizable Dataset Generation via Simulation." In Proceedings of the IEEE/CVF Conference on Computer Vision and Pattern Recognition, pp. 22401-22412. 2024.

---

> > ### Author Response · Authors · 2024-11-21
> > **Response to Reviewer V1Qj (pt 2)**
> >
> > **Q: Evaluation on unseen objects.**
> >
> > A: Thank you for your suggestion! While the focus of PHIER is on generalization to novel predicates through the inferred predicate hierarchy, we agree that further evaluation on queries with unseen objects would strengthen our paper. Hence, thanks to your suggestion, we expanded our CALVIN and BEHAVIOR experiments to evaluate accuracy on few-shot generalization on novel objects.
> >
> > |     Performance on Novel Objects |        |          |
> > |----------------------------------|--------|----------|
> > |                                  | CALVIN | BEHAVIOR |
> > | PHIER (Ours)                             | **0.851**  | **0.781**    |
> > | Re-Attention                     | 0.633  | 0.608    |
> > | CoarseFine                       | 0.562  | 0.632    |
> > | BUTD                             | 0.584  | 0.646    |
> > | RelViT                           | 0.497  | 0.642    |
> > | CLIP                             | 0.506  | 0.595    |
> > | FiLM                             | 0.411  | 0.521    |
> >
> > As in our experiments on unseen combinations and novel predicates, we observe that PHIER significantly outperforms prior baselines on unseen objects. Specifically, PHIER improves upon the top-performing prior work by 21.8 percent points on CALVIN and 13.5 percent point on BEHAVIOR. These results demonstrate that PHIER improves generalization to both novel objects and predicates, further highlighting the benefit of our object-centric encoder and inferred predicate hierarchy. We have updated Appendix Section A.5 with these new queries and additional results accordingly.
> >
> > **Q: Evaluation on GPT-4v on the real-world benchmark.**
> >
> > A: Thank you for pointing this out. We note that pre-trained models are typically trained on large-scale real-world datasets with vast amounts of diverse data (e.g., from our comparisons, BLIP-2 was trained on 129M images, ViperGPT composes several large, pre-trained models such as GLIP [1] and X-VLM [2], and GPT-4v was trained on an internal unspecified large-scale dataset). Hence, these models inherently do not differentiate between in-distribution and out-of-distribution scenarios in the real world. However, we agree that reporting results of the pre-trained models on real-world samples is more comprehensive. Thanks to your suggestion, we added these results, while noting that we consider these models our upper bound (due to their immense real-world train data), compared to PHIER and other supervised works. We compare the pre-trained models against PHIER tested under zero-shot and few-shot settings. In the zero-shot setting, we see that PHIER outperforms ViperGPT and BLIP-2 by 6.0% and 1.4% respectively, showing the potential for a small model trained on significantly less data, to reach the performance level of large pre-trained models. However, GPT-4v outperforms PHIER by 10.4%. which we hypothesize is due to its model size and dataset scale, even compared to prior pre-trained works. In the few-shot setting using only two examples, PHIER’s performance improves significantly and narrows the gap with GPT-4v to just 0.9%. This further demonstrates that PHIER’s inferred predicate hierarchy enables it to generalize to novel queries with efficient adaptation.
> >
> > |               | All   | Unseen Comb. | Novel Pred. |
> > |---------------|-------|--------------|-------------|
> > | Ours          | 0.608 | 0.632        | 0.585       |
> > | Ours (2-shot) | 0.703 | 0.714        | 0.691       |
> > | GPT-4V        | 0.712 | 0.737        | 0.688       |
> > | BLIP-2        | 0.594 | 0.591        | 0.597       |
> > | ViperGPT      | 0.548 | 0.538        | 0.557       |
> >
> > Thanks to your feedback, we have included these experiment results in Section 5 in the main text.
> >
> > [1] Li, Liunian Harold, Pengchuan Zhang, Haotian Zhang, Jianwei Yang, Chunyuan Li, Yiwu Zhong, Lijuan Wang et al. "Grounded Language-Image Pre-Training." In Proceedings of the IEEE/CVF Conference on Computer Vision and Pattern Recognition, pp. 10965-10975. 2022.
> >
> > [2] Zeng, Yan, Xinsong Zhang, and Hang Li. "Multi-Grained Vision Language Pre-Training: Aligning Texts with Visual Concepts." arXiv preprint arXiv:2111.08276 (2021)

---

> > > ### Author Response · Authors · 2024-11-21
> > > **Response to Reviewer V1Qj (pt 3)**
> > >
> > > **Q: Object-centric encoder for environments with significant distribution shifts.**
> > >
> > > A: We see empirically that PHIER’s object-centric encoder performs well even on environments with significant distribution shifts, such as CALVIN. In Figure 8 in the Appendix, we add an example of how the encoder localizes objects in CALVIN. To adapt to environments with even larger distribution shifts where the performance may decrease, we note that PHIER’s object-centric encoder can be finetuned with more data as well. We have included this example and discussion in Appendix Section A.8.
> > >
> > > **Q: Additional visualizations of the learned Poincare Ball.**
> > >
> > > A: Thank you for your suggestion. We added visualizations of the joint image-predicate space for BEHAVIOR on the Poincaré disk in Figure 6 in the Appendix, highlighting the hierarchical semantic structure captured by PHIER's embeddings. By grouping the joint image-predicate embeddings by predicate, we uncover the inferred predicate hierarchy. For instance, we see that embeddings for NextTo are positioned closer to the origin compared to those for OnLeft, accurately reflecting their hierarchical relationship — OnLeft is a more specific case of NextTo. Furthermore, embeddings for Touching are nearest to the origin, consistent with its role as the most general predicate. For example, when one object is Inside or OnTop of another, they are inherently Touching. Similarly, objects that are NextTo or OnLeft are also frequently Touching. This visualization demonstrates that PHIER captures not only semantic structure but also nuanced hierarchical relationships between predicates.
> > >
> > > We further analyze the embeddings for novel predicates after few-shot learning with only five examples. Notably, even with such limited data, PHIER successfully integrates these novel predicates into the latent space and aligns them with their learned counterparts in semantically consistent regions (e.g., OnRight is near OnLeft). By aligning these predicates in similar regions, PHIER is able to leverage its existing knowledge of relevant features for learned predicates (e.g., OnLeft) to reason about novel predicates (e.g., OnRight). This alignment highlights that PHIER effectively encodes the relationships between pairwise predicates in the latent space, enabling generalization to novel predicates with minimal examples. Thanks to your feedback, we have updated Appendix Section A.6 with these visualizations and analyses.
> > >
> > > **Q: Clarification on ablation details.**
> > >
> > > A: Thank you for pointing this out! To address your concern, we provide a clear breakdown of our ablation model architectures and explain how we add each component.
> > >
> > > Supervised model: We start with a supervised baseline model, which uses an image encoder and text incoder initialized with CLIP and BERT weights, respectively.The embeddings from both encoders are concatenated and passed through a small MLP with three linear layers for classification, and the full model is trained with a binary cross-entropy loss based on the ground truth labels (True or False). We then progressively add each component of PHIER.
> > >
> > > \+ Object-centric encoder: First, we incorporate the object-centric encoder by replacing the image encoder, text encoder, and concatenation step with our proposed object-centric encoder, while retaining the MLP and loss.
> > >
> > > \+ Predicate triplet loss: Next, we introduce the predicate triplet loss by adding this term to the total loss function without changing the architecture.
> > >
> > > \+ Norm regularization loss: We further add the norm regularization loss to get the total loss function with all components, as described in Section 3.4.
> > >
> > > \+ Hyperbolic metric: Finally, we lift the scene representation to hyperbolic space using an exponential map and replace the first two linear layers in the MLP with two hyperbolic linear layers of the same size. We also use the Poincaré distance metric instead of the Euclidean metric in the self-supervised losses, yielding our final model (PHIER).
> > >
> > > Thanks to your feedback, we have added these details about the ablation architectures in Appendix Section C.3. We also added new ablations that remove each component separately in Appendix Section A.4.

---

> > > > ### Author Response · Authors · 2024-11-21
> > > > **Response to Reviewer V1Qj (pt 4)**
> > > >
> > > > **Q: Clarification on network architecture.**
> > > >
> > > > A: Thank you for bringing this to our attention. We clarify that all of the parameters in our architecture are trainable. The image and text encoders are initialized with pre-trained CLIP and BERT weights, respectively. The hyperbolic linear layers are initialized following the approach of Shimizu et al. [1], with the weights drawn from a normal distribution centered at zero with a standard deviation $(2nm)^{-\frac{1}{2}}$, where $m$ and $n$ are the input and output sizes of the layer, and the biases set to the zero vector. The linear layer in the small MLP is initialized by the standard Kaiming initialization. These are all updated during training. We have clarified these details in Appendix Section A.1.
> > > >
> > > > [1] Shimizu, Ryohei, Yusuke Mukuta, and Tatsuya Harada. "Hyperbolic Neural Networks++." arXiv preprint arXiv:2006.08210 (2020).
> > > >
> > > > **Q: Clarification of object-centric encoder.**
> > > >
> > > > A: We would like to clarify that the object-centric image encoder is a part of PHIER’s contribution. Specifically, our method disentangles the conditioning of the image on the full state classification query into two distinct ones: one that identifies the relevant objects and another that focuses on key features for the given predicate. While we use MaskCLIP to identify the relevant entities, our primary contribution lies in the decomposition of the query into object and predicate components, enabling PHIER to faithfully identify the relevant entities and extract features based on the predicate. Thanks to your feedback, we have updated Appendix Section A.1 in the main text to further highlight our contribution.
> > > >
> > > > **Q: Choice of encoder.**
> > > >
> > > > A: We choose the BERT encoder as it is a state-of-the-art model for text understanding, and  well-suited for capturing nuanced textual semantics, such as predicate relationships, required by our task. While the CLIP text encoder is effective for vision-language alignment, it focuses on mapping text to image spaces, hence making it a less suitable choice.
> > > >
> > > > **Q: Sensitivity of model to hyperparameters.**
> > > >
> > > > A: Although balancing the self-supervised losses relative to the supervised loss is important for achieving strong performance, our model’s performance is robust and not highly sensitive to the exact values of these hyperparameters. We empirically found that across both CALVIN and BEHAVIOR, a smaller loss weight that capped the maximum loss at 0.5 led to similar performance, which we set to be 0.05 for the predicate triplet loss and 1.0 for the norm regularization loss.
> > > >
> > > > **Q: Efficiency of querying LLM.**
> > > >
> > > > A: We query the LLM once before training starts to retrieve the predicate triplet pairs and hierarchy, hence training is not affected. We explain this process and prompt in Appendix Section D.
> > > >
> > > > **Q: Clarification on GPT-4v reasoning.**
> > > >
> > > > A: We experimented with different prompting strategies to optimize GPT-4v's performance. Given that the questions in our task are relatively simple, we found that GPT-4v was more accurate and coherent when it generated an answer first, followed by its reasoning.
> > > >
> > > > **Q: ID evaluation.**
> > > >
> > > > A: Thanks for the feedback. We clarify that ID evaluation is done before few-shot training (as no adaptation is required), while OOD evaluation is done after.

---

> > > > > ### Comment · Reviewer_V1Qj · 2024-11-25
> > > > >
> > > > > Thank you for addressing my concerns.
> > > > >
> > > > > The real-world few-shot transfer results are convincing and the authors show that PHIER performs comparable to GPT4V.
> > > > > I agree that the authors method does not need to outperform GPT4V, but including the results definitely helps to better understand PHIER's perfomance.
> > > > >
> > > > > The additional real-world BEHAVIOUR benchmark further support the claims of the work. However, while the benchmark includes distractors, the setup is still rather simple in most cases. It would be interesting to see how the method performs in more challenging scenarios.
> > > > >
> > > > > Also, you state that BEHAVIOR Vision Suite is a real-world benchmark. From my understanding, it is a simulated benchmark. Could you please clarify?
> > > > > Thank you.

---

> > > > > > ### Author Response · Authors · 2024-11-25
> > > > > > **Response to Reviewer V1Qj**
> > > > > >
> > > > > > Dear Reviewer V1Qj,
> > > > > >
> > > > > > Thank you for your additional feedback! We clarify that BEHAVIOR Vision Suite [1] includes a complementary dataset of real-world images, which were collected to validate that the simulated data supports transfer learning to real-world scenarios (described in Section 4.3 of [1]). We evaluate PHIER on this complex real-world dataset.
> > > > > >
> > > > > > [1] Ge, Yunhao, Yihe Tang, Jiashu Xu, Cem Gokmen, Chengshu Li, Wensi Ai, Benjamin Jose Martinez et al. “BEHAVIOR Vision Suite: Customizable Dataset Generation via Simulation.” In Proceedings of the IEEE/CVF Conference on Computer Vision and Pattern Recognition, pp. 22401-22412. 2024.

---

### Official Review · Reviewer_vCxY · 2024-10-31

**Soundness:** 3
**Presentation:** 4
**Contribution:** 3
**Rating:** 6
**Confidence:** 3

**Summary:**

This work suggests a novel method for state classification with enhanced few-shot generalization capabilities. The method encodes latent features extracted from an Image and a query (in the form of an object conditioned predicate) onto a hyperbolic latent space using a unique loss and regularizer to enforce explicit learning of predicate hierarchies. The authors leverage the natural hierarchical structure that emerges in hyperbolic space based on previously seen predicates in the hierarchy to quickly learn and adapt to new, unseen objects and predicates.

**Strengths:**

- All deep learning components and encoding strategies are based on strong intuitions and deep understanding of the different latent spaces and tools used.
- Empirical results show significant improvement upon current state of the art.
- Tested on environments with varying levels of complexity and realism.
- Well cited with a comprehensive literature review in the related work section
- The method is clearly presented bit by bit, slowly building up the method components in a readable fashion.
- Provides ablation tests that show the necessity of each component.

**Weaknesses:**

- Theoretical justification of the method is relatively weak. It would be groundbreaking to show a strongly linked connection between hyperbolic space and hierarchical structure. This point is still unclear, even though the authors provide some intuition for it.
- Introduction lacks citations, making it difficult to understand which parts are novel and which belong to previous work (before diving in to the rest of the paper).
- No evidence is provided for claims on that large vision models struggle with spatial relationships, nor is there mention of multi-perspective image processing with these kinds of models (which emulates 3D and improves spatial understanding).
- The authors assume much preliminary knowledge from the reader, e.g., acronyms like CLIP, BERT, ViT, etc., are never defined. Even though these are common terms in the deep learning and NLP communities, Some readers might still not be familiar with everything which requires reference hopping and a general breaking of the reading flow.
- tested on very small datasets which makes the results less statistically significance.

**Questions:**

- in line 228 the authors define the image encoding with a max over all object mask norms. doesn't this cause the final embedding to ignore the objects whose norm
- shouldn't the margin hyperparameter (lambda) be chosen dynamically depending on the current triplet? Otherwise this encourages a minimal margin that lead to equal distances between values that have significantly different semantic meaning.
- Why does the hyperbolic space's disc area growing exponentially mean that it is a continuous representation of discrete trees? There seems to be no connection between trees and the hyperbolic space according to the information provided in this work.

---

> ### Author Response · Authors · 2024-11-21
> **Response to  Reviewer vCxY (pt1)**
>
> We thank you for the constructive feedback!
>
> **Q: Evaluation on a larger real-world dataset.**
>
> A: Thank you for this suggestion! We have added additional evaluation on BEHAVIOR Vision Suite [1], a new and more complex real-world benchmark. The dataset consists of 500 examples with significantly more diverse scenes and distractor objects. Specifically, compared to our train data, this benchmark includes 10 unseen combinations (171 examples), 10 novel predicates (166 examples), and 10 novel objects (163 examples). See dataset examples in the updated Figure 3 in the main text as well as in Figure 12 in the Appendix.
> We present the zero- and few-shot real-world transfer results of PHIER and previous supervised models trained on the simulated BEHAVIOR dataset, and then tested on this real-world dataset. For a comprehensive comparison, we also evaluate pre-trained models as our upper bound.
> |              | Zero-Shot |        |            | 2-Shot |        |            |
> |--------------|-----------|--------|------------|--------|--------|------------|
> |              | All       | Unseen Comb. | Novel Pred. | All    | Unseen Comb. | Novel Pred. |
> | Ours         | 0.608     | 0.632  | 0.585      | 0.703  | 0.714  | 0.691      |
> | Re-Attention | 0.377     | 0.415  | 0.341      | 0.413  | 0.458  | 0.368      |
> | CoarseFine   | 0.490      | 0.485  | 0.494      | 0.553  | 0.562  | 0.543      |
> | BUTD         | 0.418     | 0.427  | 0.409      | 0.456  | 0.464  | 0.448      |
> | RelViT       | 0.553     | 0.579  | 0.528      | 0.603  | 0.654  | 0.552      |
> | CLIP         | 0.516     | 0.544  | 0.489      | 0.571  | 0.674  | 0.468      |
> | FiLM         | 0.459     | 0.480   | 0.438      | 0.513  | 0.542  | 0.484      |
> | GPT-4V       | 0.712     | 0.737  | 0.688      | --     | --     | --         |
> | BLIP-2       | 0.599     | 0.602  | 0.597      | --     | --     | --         |
> | ViperGPT     | 0.553     | 0.538  | 0.568      | --     | --     | --         |
>
> We observe similar trends as in our manually-collected real-world setup, with PHIER significantly outperforming prior supervised baselines on this challenging sim-to-real task across both zero- and few-shot settings. We conjecture that this is because PHIER learns more robust features for images—only features core to the specified state classification task are captured, and hence enables PHIER to generalize and remain invariant to the visual details in the real world. However, as expected, pre-trained models trained on large-scale real-world data outperform PHIER due to their large training data corpus. We added experiment results and examples of the dataset in Section 5 of the main text and Appendix Section A.2.
>
> [1] Ge, Yunhao, Yihe Tang, Jiashu Xu, Cem Gokmen, Chengshu Li, Wensi Ai, Benjamin Jose Martinez et al. "BEHAVIOR Vision Suite: Customizable Dataset Generation via Simulation." In Proceedings of the IEEE/CVF Conference on Computer Vision and Pattern Recognition, pp. 22401-22412. 2024.

---

> > ### Author Response · Authors · 2024-11-21
> > **Response to Reviewer vCxY (pt2)**
> >
> > **Q: Justification of connection between hyperbolic space and hierarchical structure.**
> >
> > A: We highlight several prominent prior works who have made theoretical connections between hyperbolic space and trees. Mathematical works such as Gromov [1], Dyubina and Polterovich [2], and Hamann [3] prove that any finite tree can be embedded into a finite hyperbolic space with approximately preserved distances. A key property of hyperbolic space is its exponentially growing distance, and they show that this underlying property makes hyperbolic space well-suited to model hierarchical structures. Furthermore, works such as Sala et al. [4] and Chami et al. [5] propose concrete approaches to embed any tree in hyperbolic space with arbitrarily low distortion, establishing upper upper and lower bounds for distortion and further demonstrating the effectiveness of hyperbolic space for hierarchical modeling.
> >
> > Notably, Nickel and Kiela [6] were among the first to explore learning hierarchical representations in hyperbolic space. They found that for data with latent hierarchies, embeddings on the Poincaré ball outperform Euclidean embeddings significantly in terms of representation capacity and generalization ability. Since then, hyperbolic spaces have been increasingly explored for modeling hierarchies across various domains, including NLP [7, 8, 9] and computer vision [10, 11], with substantial empirical evidence supporting its efficiency and suitability for modeling hierarchical structures in comparison to Euclidean space. We believe that these prior works provide strong theoretical justification and empirical support for the connection between hyperbolic space and hierarchical structure, which inspires our method. We have included these discussions in Appendix Section C.
> >
> > [1] Gromov, Mikhael. "Hyperbolic Groups." Essays in Group Theory. New York, NY: Springer New York, 1987. 75-263.
> >
> > [2] Dyubina, Anna, and Iosif Polterovich. "Explicit Constructions of Universal ℝ-Trees and Asymptotic Geometry of Hyperbolic Spaces." Bulletin of the London Mathematical Society 33.6 (2001): 727-734.
> >
> > [3] Hamann, Matthias. "On the Tree-Likeness of Hyperbolic Spaces." In Mathematical Proceedings of the Cambridge Philosophical Society, vol. 164, no. 2, pp. 345-361. Cambridge University Press, 2018.
> >
> > [4] Sala, Frederic, Chris De Sa, Albert Gu, and Christopher Ré. "Representation Tradeoffs for Hyperbolic Embeddings." In International Conference on Machine Learning, pp. 4460-4469. PMLR, 2018.
> >
> > [5] Chami, Ines, Albert Gu, Vaggos Chatziafratis, and Christopher Ré. "From Trees to Continuous Embeddings and Back: Hyperbolic Hierarchical Clustering." Advances in Neural Information Processing Systems 33 (2020).
> >
> > [6] Nickel, Maximillian, and Douwe Kiela. "Poincaré Embeddings for Learning Hierarchical Representations." Advances in Neural Information Processing Systems 30 (2017).
> >
> > [7] ​​Ganea, Octavian, Gary Bécigneul, and Thomas Hofmann. "Hyperbolic Neural Networks." Advances in Neural Information Processing Systems 31 (2018).
> >
> > [8] Nickel, Maximillian, and Douwe Kiela. "Learning Continuous Hierarchies in the Lorentz Model of Hyperbolic Geometry." In International Conference on Machine Learning, pp. 3779-3788. PMLR, 2018.
> >
> > [9] Tifrea, Alexandru, Gary Bécigneul, and Octavian-Eugen Ganea. "Poincar\'e Glove: Hyperbolic Word Embeddings." arXiv preprint arXiv:1810.06546 (2018).
> >
> > [10] Khrulkov, Valentin, Leyla Mirvakhabova, Evgeniya Ustinova, Ivan Oseledets, and Victor Lempitsky. "Hyperbolic Image Embeddings." In Proceedings of the IEEE/CVF Conference on Computer Vision and Pattern Recognition, pp. 6418-6428. 2020.
> >
> > [11] Ermolov, Aleksandr, Leyla Mirvakhabova, Valentin Khrulkov, Nicu Sebe, and Ivan Oseledets. "Hyperbolic Vision Transformers: Combining Improvements in Metric Learning." In Proceedings of the IEEE/CVF Conference on Computer Vision and Pattern Recognition, pp. 7409-7419. 2022.

---

> > > ### Author Response · Authors · 2024-11-21
> > > **Response to Reviewer vCxY (pt3)**
> > >
> > > **Q: Hyperbolic space’s disc area.**
> > >
> > > A:  We provide further details on why the exponential growth of the disc area in hyperbolic space provides a natural and efficient way to represent trees. Note that for a regular tree with a constant branching factor $b$, the number of nodes increases exponentially with the distance from the root, as $(b+1)b^{\mathcal{l}-1}$. We can embed trees in hyperbolic space, as they mirror this exponential growth. For instance, in a two-dimensional hyperbolic space with constant curvature $K = -1$, the circumference of a disc with radius $r$ is $2\pi\sinh r$ while the area of a disc is $2\pi(\cosh r - 1)$. Since $\sinh r = \frac{1}{2} (e^r - e^{-r})$ and $cosh r = \frac{1}{2} (e^r + e^{-r})$, both the circumference and area of the disc grow exponentially with the radius.
> > >
> > > This exponential growth allows us to efficiently embed tree structures in hyperbolic space: nodes that are $\mathcal{l}$ levels from the root can be placed on the hyperbolic disc with a radius proportional to its level $\mathcal{l}$, while nodes less than $\mathcal{l}$ levels within the sphere. Thus, we see how this property allows hyperbolic space to serve as a continuous representation of discrete trees. Thanks to your feedback, we have added these details to Appendix Section C for further clarification.
> > >
> > > [1] Gromov, Mikhael. "Hyperbolic Groups." Essays in Group Theory. New York, NY: Springer New York, 1987. 75-263.
> > >
> > > [2] Dyubina, Anna, and Iosif Polterovich. "Explicit Constructions of Universal ℝ-Trees and Asymptotic Geometry of Hyperbolic Spaces." Bulletin of the London Mathematical Society 33.6 (2001): 727-734.
> > >
> > > [3] Nickel, Maximillian, and Douwe Kiela. "Poincaré embeddings for learning hierarchical representations." Advances in Neural Information Processing Systems 30 (2017).
> > >
> > > [4] Chami, Ines, Albert Gu, Vaggos Chatziafratis, and Christopher Ré. "From Trees to Continuous Embeddings and Back: Hyperbolic Hierarchical Clustering." Advances in Neural Information Processing Systems 33 (2020): 15065-15076.
> > >
> > > **Q: Evidence that large vision models struggle with spatial relationships.**
> > >
> > > A: We highlight several recent works which have explored the limitations of large vision models (VLMs) in spatial understanding. Notable examples include Liu et al. [1], which systematically analyzes the weaknesses of VLMs in spatial reasoning tasks, and Tong et al. [2], which demonstrates that multimodal LLMs often fail to encode spatial relationships effectively. Additionally, this remains an active area of research, as explored in works like Chen et al. [3], Fu et al. [4], and Cheng et al. [5]. In our paper, results on CALVIN and BEHAVIOR benchmarks further support these findings, showing significant room for improvement in understanding these predicate relationships.
> > >
> > > We also note our task takes a single image as input, while methods that conduct multi-perspective image processing generally require multiple viewpoints. Methods that generate a full explicit 3D scene from a 2D image such as Sargent et al. [6] takes hours to lift images into 3D, and still require subsequent object detection and 3D classifier training for predicate classification. Moreover, these methods struggle on scenes that contain multiple objects, which many of the scenes in our test sets contain.
> > >
> > > [1] Liu, Fangyu, Guy Emerson, and Nigel Collier. "Visual Spatial Reasoning." Transactions of the Association for Computational Linguistics 11 (2023): 635-651.
> > >
> > > [2] Tong, Shengbang, Zhuang Liu, Yuexiang Zhai, Yi Ma, Yann LeCun, and Saining Xie. "Eyes Wide Shut? Exploring the Visual Shortcomings of Multimodal LLMs." In Proceedings of the IEEE/CVF Conference on Computer Vision and Pattern Recognition, pp. 9568-9578. 2024.
> > >
> > > [3] Chen, Boyuan, Zhuo Xu, Sean Kirmani, Brain Ichter, Dorsa Sadigh, Leonidas Guibas, and Fei Xia. "SpatialVLM: Endowing Vision-Language Models with Spatial Reasoning Capabilities." In Proceedings of the IEEE/CVF Conference on Computer Vision and Pattern Recognition, pp. 14455-14465. 2024.
> > >
> > > [4] Fu, Xingyu, Yushi Hu, Bangzheng Li, Yu Feng, Haoyu Wang, Xudong Lin, Dan Roth, Noah A. Smith, Wei-Chiu Ma, and Ranjay Krishna. "Blink: Multimodal Large Language Models Can See But Not Perceive." In European Conference on Computer Vision, pp. 148-166. Springer, Cham, 2025.
> > >
> > > [5] Cheng, An-Chieh, Hongxu Yin, Yang Fu, Qiushan Guo, Ruihan Yang, Jan Kautz, Xiaolong Wang, and Sifei Liu. "SpatialRGPT: Grounded Spatial Reasoning in Vision Language Model." arXiv preprint arXiv:2406.01584 (2024).
> > >
> > > [6] Sargent, Kyle, Zizhang Li, Tanmay Shah, Charles Herrmann, Hong-Xing Yu, Yunzhi Zhang, Eric Ryan Chan et al. "ZeroNVS: Zero-Shot 360-Degree View Synthesis from a Single Image." In Proceedings of the IEEE/CVF Conference on Computer Vision and Pattern Recognition, pp. 9420-9429. 2024.

---

> > > > ### Author Response · Authors · 2024-11-21
> > > > **Response to Reviewer vCxY (pt4)**
> > > >
> > > > **Q: Clarification of preliminary knowledge.**
> > > >
> > > > A: Thanks for pointing this out! We clarify acronyms throughout the paper, especially in Section 2 and 3 of the main text.
> > > >
> > > > **Q: Clarification of image encoding.**
> > > >
> > > > A: We assume the reviewer is asking about whether the final embedding ignores objects whose norms are smaller. We clarify that this is precisely how PHIER retrieves object-centric encodings of the image; for cases where the predicate is binary and our model needs to attend to more than one object, we extract a mask for each object in the query.
> > > >
> > > > **Q: Choice of the margin hyperparameter.**
> > > >
> > > > A: While we agree that dynamically adjusting the margin hyperparameter could potentially lead to greater separation in the latent space and is an interesting direction to explore, we found that a constant margin was sufficient to encourage meaningful semantic structure in the latent space. The uniform sampling of triplets allows the fixed margin to effectively distinguish values with significantly different semantic meanings, and we believe that the additional complexity of a dynamic margin is not necessary to achieve robust semantic separation. To illustrate this, we have added visualizations and analyses of PHIER’s learned image-predicate latent space in the Appendix Section A.6, which demonstrate how our approach successfully maintains semantic distinctions.
> > > >
> > > > **Q: Citations in introduction.**
> > > >
> > > > A: Thank you for the suggestion. We have updated the paper with citations in the introduction in Section 1, and clarified our contribution.

---

> > > > > ### Author Response · Authors · 2024-11-29
> > > > > **Happy to answer additional questions**
> > > > >
> > > > > Dear Reviewer vCxY, thank you once again for your valuable feedback which has helped us greatly improve the quality of our paper. We’ve addressed your questions in our earlier response and provided new results. As the end of the discussion period is approaching, we want to make sure we have addressed all of your concerns. Please let us know if you have any additional comments. Thank you!

---

### Official Review · Reviewer_5wzn · 2024-11-04

**Soundness:** 2
**Presentation:** 2
**Contribution:** 2
**Rating:** 6
**Confidence:** 4

**Summary:**

The paper proposes PHIER, a state classification model that leverages predicate hierarchies to achieve few-shot and out-of-distribution generalization for robotic state classification tasks. PHIER incorporates an object-centric encoder, self-supervised losses to infer semantic relationships between predicates, and a hyperbolic distance metric to encode hierarchical structures. The approach is tested in simulated environments (CALVIN, BEHAVIOR) and in real-world settings.

**Strengths:**

1. Incorporates predicate hierarchies effectively for few-shot state classification.
2. Demonstrates strong generalization from simulator images to real-world images.
3. Utilizes hyperbolic space to encode complex hierarchical relationships efficiently.

**Weaknesses:**

1. Why does the proposed method perform worse than baseline methods on in-distribution samples? More detailed analysis and insights are needed.
2. Why are there no results for pre-trained models on ID-OOD samples or real-world samples?
3. In the ablation study, components are added sequentially; however, the order of addition matters. Further analysis with ablations that remove each component should be conducted.
4. Important details are missing, such as the process for constructing positive and negative pairs and how the predicate hierarchy is built.
5. The experimental evaluation requires improvement. While the collected real-world dataset is valuable, the sample size (100) is too small, and the scenario settings lack diversity, which may lead to overfitting. There are numerous existing VQA datasets that include object relations and state information. The proposed method should be evaluated on these larger and more diverse benchmarks.
6. The proposed method claims to generalize to unseen object-predicate combinations and novel predicates with few examples. However, the specific number of examples used is unclear, as it is not mentioned in the paper or the appendix. And is there an ablation study analyzing the effect of varying the number of examples?

**Questions:**

Here are questions that align with the identified limitations:

1. What factors contribute to the proposed method's lower performance compared to baseline methods on in-distribution samples? Could more detailed analysis and insights be provided?
2. Why were pre-trained models not evaluated on ID-OOD samples or real-world samples, and what impact would their inclusion have on the results?
3. In the ablation study, why were the components added sequentially without considering the effect of different orderings? Has an ablation analysis been conducted that removes individual components to assess their independent impact?
4. How were the positive and negative pairs constructed, and what methodology was used to build the predicate hierarchy? Can more detailed explanations be provided?
5. Given the small sample size (100) and limited diversity of the collected real-world dataset, how can the proposed method avoid overfitting to this specific setup? Why was it not evaluated on larger, more diverse VQA datasets that include object relations and state information?
6. How many examples are needed for the proposed method to generalize effectively to unseen object-predicate combinations and novel predicates? Is there an ablation study examining the effect of varying the number of examples?

---

> ### Author Response · Authors · 2024-11-21
> **Response to Reviewer 5wzn (pt1)**
>
> We thank you for the constructive feedback!
>
> **Q: Evaluation on a larger real-world dataset.**
>
> A: Thank you for this suggestion! We have added additional evaluation on BEHAVIOR Vision Suite [1], a new and more complex real-world benchmark. The dataset consists of 500 examples with significantly more diverse scenes and distractor objects. Specifically, compared to our train data, this benchmark includes 10 unseen combinations (171 examples), 10 novel predicates (166 examples), and 10 novel objects (163 examples). See dataset examples in the updated Figure 3 in the main text as well as in Figure 12 in the Appendix.
> We present the zero- and few-shot real-world transfer results of PHIER and previous supervised models trained on the simulated BEHAVIOR dataset, and then tested on this real-world dataset. For a comprehensive comparison, we also evaluate pre-trained models as our upper bound.
> |              | Zero-Shot |        |            | 2-Shot |        |            |
> |--------------|-----------|--------|------------|--------|--------|------------|
> |              | All       | Unseen Comb. | Novel Pred. | All    | Unseen Comb. | Novel Pred. |
> | Ours         | 0.608     | 0.632  | 0.585      | 0.703  | 0.714  | 0.691      |
> | Re-Attention | 0.377     | 0.415  | 0.341      | 0.413  | 0.458  | 0.368      |
> | CoarseFine   | 0.490      | 0.485  | 0.494      | 0.553  | 0.562  | 0.543      |
> | BUTD         | 0.418     | 0.427  | 0.409      | 0.456  | 0.464  | 0.448      |
> | RelViT       | 0.553     | 0.579  | 0.528      | 0.603  | 0.654  | 0.552      |
> | CLIP         | 0.516     | 0.544  | 0.489      | 0.571  | 0.674  | 0.468      |
> | FiLM         | 0.459     | 0.480   | 0.438      | 0.513  | 0.542  | 0.484      |
> | GPT-4V       | 0.712     | 0.737  | 0.688      | --     | --     | --         |
> | BLIP-2       | 0.599     | 0.602  | 0.597      | --     | --     | --         |
> | ViperGPT     | 0.553     | 0.538  | 0.568      | --     | --     | --         |
>
> We observe similar trends as in our manually-collected real-world setup, with PHIER significantly outperforming prior supervised baselines on this challenging sim-to-real task across both zero- and few-shot settings. We conjecture that this is because PHIER learns more robust features for images—only features core to the specified state classification task are captured, and hence enables PHIER to generalize and remain invariant to the visual details in the real world. However, as expected, pre-trained models trained on large-scale real-world data outperform PHIER due to their large training data corpus. We added experiment results and examples of the dataset in Section 5 of the main text and Appendix Section A.2.
>
> [1] Ge, Yunhao, Yihe Tang, Jiashu Xu, Cem Gokmen, Chengshu Li, Wensi Ai, Benjamin Jose Martinez et al. "BEHAVIOR Vision Suite: Customizable Dataset Generation via Simulation." In Proceedings of the IEEE/CVF Conference on Computer Vision and Pattern Recognition, pp. 22401-22412. 2024.

---

> > ### Author Response · Authors · 2024-11-21
> > **Response to Reviewer 5wzn (pt2)**
> >
> > **Q: Ablation study on number of examples needed to generalize.**
> >
> > A: Thank you for the suggestion. We clarify that the methods use 5 examples for few-shot generalization (414-415), and we have better highlighted this in Section 5 of the main text. In addition, we agree that ablations studying the effect of varying the number of examples used is important. Based on your feedback, we added new ablation experiments with 0, 1, 2, 3, 4, 5, and 10-shot generalization performance on both CALVIN and BEHAVIOR environments. We show plots of the results in Figure 5 in the Appendix.
> >
> > |              | CALVIN |         |       |       |       |       |       | BEHAVIOR       |       |       |       |       |       |         |
> > |--------------|--------|----------|-------|-------|-------|-------|-------|-------|-------|-------|-------|-------|-------|---------|
> > |   n-shot           | 0      | 1        | 2     | 3     | 4     | 5     | 10    | 0     | 1     | 2     | 3     | 4     | 5     | 10      |
> > | Ours         | **0.559** | **0.713**    | **0.792** | **0.829** | **0.854** | **0.899** | **0.907** | **0.651** | **0.693** | **0.722** | **0.765** | **0.783** | **0.820**  | **0.841**   |
> > | Re-Attention | 0.442  | 0.519    | 0.542 | 0.571 | 0.607 | 0.674 | 0.729 | 0.422 | 0.479 | 0.512 | 0.556 | 0.599 | 0.652 | 0.733   |
> > | CoarseFine   | 0.423  | 0.508    | 0.531 | 0.544 | 0.582 | 0.624 | 0.650  | 0.506 | 0.522 | 0.552 | 0.583 | 0.613 | 0.636 | 0.696   |
> > | BUTD         | 0.405  | 0.502    | 0.531 | 0.558 | 0.572 | 0.585 | 0.621 | 0.541 | 0.602 | 0.641 | 0.675 | 0.683 | 0.712 | 0.751   |
> > | RelViT       | 0.371  | 0.453    | 0.481 | 0.512 | 0.532 | 0.563 | 0.601 | 0.478 | 0.536 | 0.584 | 0.635 | 0.681 | 0.737 | 0.767   |
> > | CLIP         | 0.282  | 0.395    | 0.446 | 0.485 | 0.501 | 0.546 | 0.589 | 0.385 | 0.451 | 0.512 | 0.563 | 0.580  | 0.632 | 0.680    |
> > | FiLM         | 0.236  | 0.340     | 0.374 | 0.415 | 0.452 | 0.489 | 0.542 | 0.412 | 0.480  | 0.510  | 0.541 | 0.570  | 0.583 | 0.620    |
> >
> > The results show that PHIER consistently outperforms prior works across all numbers of examples. Notably, in the CALVIN environment, PHIER’s performance plateaus as the number of examples increases, indicating that the method requires only a few examples to adapt effectively to unseen scenarios. We have included these experiment results in Appendix Section A.3.

---

> > > ### Author Response · Authors · 2024-11-21
> > > **Response to Reviewer 5wzn (pt3)**
> > >
> > > **Q: Evaluation on ID-OOD and real world samples.**
> > >
> > > A: Thank you for the suggestion. We note that pre-trained models are typically trained on large-scale real-world datasets with vast amounts of diverse data (e.g., from our comparisons, BLIP-2 was trained on 129M images, ViperGPT composes several large, pre-trained models such as GLIP [1] and X-VLM [2], and GPT-4v was trained on an internal unspecified large-scale dataset). Hence, these models inherently do not differentiate between ID and OOD scenarios, as their training data overlaps significantly with both. We agree that reporting results on ID-OOD is more comprehensive, hence thanks to your suggestion, we added results for pre-trained models on ID-OOD. We see that pre-trained models demonstrate similar performance ID and OOD, and PHIER still significantly outperforms all models (e.g., by 33.6% on OOD CALVIN and 11.4% on OOD BEHAVIOR). These results reveal no consistent trend between ID and OOD scenarios for pre-trained models. Notably, the performance drop for PHIER between ID and OOD is comparable to that of the pre-trained models, demonstrating that our model's generalization behavior is within a reasonable range.
> > >
> > > |     ID-OOD|        |           |
> > > |--------------------|--------|-----------|
> > > |              | CALVIN | BEHAVIOR  |
> > > | PHIER (Ours)               | 0.046  | 0.039     |
> > > | GPT-4v             | 0.024  | -0.045    |
> > > | BLIP-2             | -0.003 | 0.019     |
> > > | ViperGPT           | -0.009 | -0.031    |
> > >
> > > Similarly, we add in additional results of the pre-trained models on real-world samples, while noting that we consider these models our upper bound (due to their immense real-world train data), compared to PHIER and other supervised works. We compare the pre-trained models against PHIER tested under zero-shot and few-shot settings. In the zero-shot setting, we see that PHIER outperforms ViperGPT and BLIP-2 by 6.0% and 1.4% respectively, showing the potential for a small model trained on significantly less data, to reach the performance level of large pre-trained models. However, GPT-4v outperforms PHIER by 10.4%. which we hypothesize is due to its model size and dataset scale, even compared to prior pre-trained works. In the few-shot setting using only two examples, PHIER’s performance improves significantly and narrows the gap with GPT-4v to just 0.9%. This further demonstrates that PHIER’s inferred predicate hierarchy enables it to generalize to novel queries with efficient adaptation.
> > >
> > > |               | All   | Unseen Comb. | Novel Pred. |
> > > |---------------|-------|--------------|-------------|
> > > | Ours          | 0.608 | 0.632        | 0.585       |
> > > | Ours (2-shot) | 0.703 | 0.714        | 0.691       |
> > > | GPT-4V        | 0.712 | 0.737        | 0.688       |
> > > | BLIP-2        | 0.594 | 0.591        | 0.597       |
> > > | ViperGPT      | 0.548 | 0.538        | 0.557       |
> > >
> > > Thanks to your feedback, we have included both of these experiment results in Section 5 in the main text.
> > >
> > > [1] Li, Liunian Harold, Pengchuan Zhang, Haotian Zhang, Jianwei Yang, Chunyuan Li, Yiwu Zhong, Lijuan Wang et al. "Grounded Language-Image Pre-Training." In Proceedings of the IEEE/CVF Conference on Computer Vision and Pattern Recognition, pp. 10965-10975. 2022.
> > >
> > > [2] Zeng, Yan, Xinsong Zhang, and Hang Li. "Multi-Grained Vision Language Pre-Training: Aligning Texts with Visual Concepts." arXiv preprint arXiv:2111.08276 (2021).

---

> > > > ### Author Response · Authors · 2024-11-21
> > > > **Response to Reviewer 5wzn (pt4)**
> > > >
> > > > **Q: Ablation study with removed components.**
> > > >
> > > > A: Thanks for bringing up this point! To address your concern, we added an ablation study that evaluates the impact of removing individual components of PHIER to evaluate their contributions.
> > > >
> > > > We compare PHIER with 4 variants, (1) without the object-centric encoder, (2) without the predicate triplet loss, (3) without the norm regularization loss, and (4) without the hyperbolic latent space. We see that without our object-centric design, performance drops significantly in both ID and OOD settings, emphasizing the importance of object-centric encoders for improved representation and reasoning. In addition, we show that removing each of the self-supervised losses leads to much weaker generalization capability. Finally, we observe reduced generalization performance without PHIER’s hyperbolic latent space and hyperbolic norm regularization loss, demonstrating that the hyperbolic space facilitates better handling of hierarchical relationships.
> > > >
> > > > |                     | CALVIN | |        |    BEHAVIOR   |      |          |
> > > > |---------------------|--------|----------|--------|-------|-------|----------|
> > > > |                     | ID     | OOD      | ID-OOD | ID    | OOD   | ID-OOD   |
> > > > | PHIER               | 0.945  | 0.899    | 0.046  | 0.859 | 0.83  | 0.029    |
> > > > | - Object centric encoder    | 0.786  | 0.704    | 0.082  | 0.703 | 0.659 | 0.044    |
> > > > | - Predicate triplet loss | 0.867  | 0.601    | 0.266  | 0.774 | 0.624 | 0.150     |
> > > > | - Norm regularization loss     | 0.914  | 0.823    | 0.091  | 0.834 | 0.782 | 0.052    |
> > > > | - Hyperbolic metric       | 0.903  | 0.784    | 0.119  | 0.803 | 0.761 | 0.042    |
> > > >
> > > > These results validate that each component contributes meaningfully to PHIER’s performance, particularly in improving OOD generalization. Thanks to your feedback, we have included these experiment results in Appendix Section A.4.
> > > >
> > > > **Q: Performance on in-distribution samples.**
> > > >
> > > > A: We note that in the in-distribution (ID) setting of CALVIN, PHIER outperforms all prior works except Re-Attention, with only a small margin of 1.4%. In the out-of-distribution (OOD) setting, which is our primary focus, PHIER outperforms Re-Attention by a significant 22.5%. Similarly, on ID BEHAVIOR, PHIER performs comparably to top-performing prior works, surpassing all except RelViT by 0.7%; however, in the OOD setting we focus on, PHIER outperforms RelViT by 8.3%. We highlight that PHIER performs comparably to top-performing prior works in the ID setting, while significantly improving the OOD performance. We focus on the few-shot generalization task and design our method to enforce bottlenecked representations (via a joint image-predicate space), while acknowledging that this might include tradeoffs on ID performance to avoid overfitting to the train distribution.
> > > >
> > > > We also analyze specific cases where PHIER underperforms on ID examples. For instance, in CALVIN, we hypothesize that PHIER may struggle with tasks that the baselines may memorize due to their less constrained representations. We add an example in Figure 7 of the Appendix, and note that for the ID query, TurnedOn(lightbulb), Re-Attention correctly predicts True, while PHIER predicts False. However, for the out-of-distribution query, TurnedOff(lightbulb), which is linguistically similar but semantically opposite, PHIER generalizes successfully while Re-Attention struggles to adapt. We conjecture that Re-Attention may predict that TurnedOn(lightbulb) is True based solely on the existence of the bulb at the location, instead of learning that the state of the lightbulb depends on its color (yellow is on and white is off). In contrast, we see that although PHIER’s constrained representation may slightly limit learning capacity for ID settings, PHIER has the potential to conduct better compositional reasoning in OOD scenarios, where PHIER significantly outperforms baselines. Thanks to your feedback, we have included a more in-depth discussion of this in Appendix Section A.7.

---

> > > > > ### Author Response · Authors · 2024-11-21
> > > > > **Response to Reviewer 5wzn (pt5)**
> > > > >
> > > > > **Q: Clarification on predicate hierarchy construction.**
> > > > >
> > > > > A: Thank you for pointing this out. At a high level, we use large language models (LLMs), specifically GPT-4, to construct the predicate hierarchy. More concretely, we first sample triplets of predicates from the data. For each triplet, we prompt the LLM to assess the underlying relationships between the predicates. One predicate in the triplet is randomly chosen as the anchor. The LLM is asked to determine which of the other two predicates is more similar to the anchor. The anchor and the more similar predicate form a positive pair, while the anchor and the less similar predicate form a negative pair. By extracting knowledge from an LLM, we leverage the LLM's explicit and extensive understanding of predicate relationships to produce meaningful triplets and guide the model toward a semantically rich image-predicate latent space. We clarify this process in Section 3 of the main text and include the prompt in Appendix Section D.

---

> > > > > > ### Author Response · Authors · 2024-11-29
> > > > > > **Happy to answer additional questions**
> > > > > >
> > > > > > Dear Reviewer 5wzn, thank you once again for your valuable feedback which has helped us greatly improve the quality of our paper. We’ve addressed your questions in our earlier response and provided new results. As the end of the discussion period is approaching, we want to make sure we have addressed all of your concerns. Please let us know if you have any additional comments. Thank you!

---

> > > > > > > ### Comment · Reviewer_5wzn · 2024-12-01
> > > > > > >
> > > > > > > Thank the authors for the detailed rebuttal. I have decided to raise my rating.

---

> > > > > > > > ### Author Response · Authors · 2024-12-02
> > > > > > > > **Thanks to Reviewer 5wzn**
> > > > > > > >
> > > > > > > > Thank you again for the thoughtful comments and feedback.

---

### Meta-Review · Area_Chair_FszK · 2024-12-19

**Metareview:**

The paper proposes a method called PHIER for few-shot state classification that encodes predicate hierarchies. The initial review identified several weaknesses such as the need for more detailed analysis on in-distribution sample performance, lack of results for pre-trained models on certain samples, issues with the ablation study, missing details in the experimental setup, and the need to clarify the number of examples for generalization.
The reviewers acknowledged the improvements made by the authors in their rebuttals. The additional real-world experiments and clarifications strengthened the paper's claims and addressed many of the initial weaknesses. Although there were some remaining minor concerns about the simplicity of the evaluation scenarios, overall, the paper now presents a more complete and convincing study with significant contributions in the area of few-shot state classification. The improvements in the paper's content and the authors' responsiveness to reviewer feedback justify the acceptance of the paper.

**Additional Comments On Reviewer Discussion:**

The authors provided comprehensive responses to all the reviewer concerns. They added evaluations on a real-world dataset (BEHAVIOR Vision Suite), conducted ablation studies on the number of examples needed for generalization and the impact of removing individual components, provided justifications for the connection between hyperbolic space and hierarchical structure, and addressed other questions regarding experimental details, model architecture, and assumptions.

---

### Decision · Program_Chairs · 2025-01-22

Accept (Poster)